# Gradient-Free Approaches is a Key to an Efficient Interaction with Markovian Stochasticity

**Boris Prokhorov** [* 1]  **Semyon Chebykin** [* 2]  **Alexander Gasnikov** [3]  **Aleksandr Beznosikov** [4 3 5]

## Abstract

This paper deals with stochastic optimization problems involving Markovian noise with a zero-order oracle. We present and analyze a novel derivative-free method for solving such problems in strongly convex smooth and non-smooth settings with both one-point and two-point feedback oracles. Using a randomized batching scheme, we show that when mixing time $\tau$ of the underlying noise sequence is less than the dimension of the problem $d$, the convergence estimates of our method do not depend on $\tau$. This observation provides an efficient way to interact with Markovian stochasticity: instead of invoking the expensive first-order oracle, one should use the zero-order oracle. Finally, we complement our upper bounds with the corresponding lower bounds. This confirms the optimality of our results.

## 1. Introduction

Stochasticity is a fundamental aspect of many optimization problems, naturally arising in the field of machine learning (Shalev-Shwartz & Ben-David, 2014; Goodfellow et al., 2016). Stochastic gradient descent (SGD) (Robbins & Monro, 1951) and its accelerated variants (Nesterov, 1983; Ghadimi & Lan, 2016) have become a de facto optimizers for modern large models training. Theoretical properties of SGD have been extensively studied under various statistical frameworks (Moulines & Bach, 2011; Ghadimi & Lan, 2013; Dieuleveut et al., 2017; Vaswani et al., 2019), often relying on the assumption that noise is independent and identically distributed (i.i.d.). However, in many real-world applications — including reinforcement learning (RL) (Bhandari et al., 2018; Durmus et al., 2021), distributed optimization (Lopes & Sayed, 2007; Johansson et al., 2007), and bandit problems (Auer, 2002) — noise is not i.i.d., instead exhibiting correlations or *Markovian structure*.

For instance, in the mentioned growing field of RL, sequential interactions with the environment induce state-dependent structure of the noise, creating a need for non-i.i.d. noise aware algorithms. Although several gradient-based methods for Markovian stochastic oracles have been studied in the past decade (Duchi et al., 2012; Even, 2023), policy optimization in RL is based solely on reward feedback, making traditional methods inapplicable, since there is no access to first-order information (Salimans et al., 2017; Choromanski et al., 2018; Fazel et al., 2018). *Zero-order optimization* (ZOO) methods are specifically developed to address such problems, and are used in scenarios where gradients are unavailable or prohibitively expensive to compute. Apart from RL, ZOO techniques are widely employed in adversarial attack generation (Chen et al., 2017), hyperparameter tuning (Shahriari et al., 2016; Wu et al., 2020), continuous bandits (Bubeck et al., 2012; Shamir, 2017) and other applications (Taskar et al., 2005; Lian et al., 2015). While the literature on ZOO is extensive, this work is, to our knowledge, *the first study of optimization problem with both zero-order information and Markovian noise*, aimed at developing an optimal algorithm for a large family of problems from the intersection of these two areas.

### 1.1. Related works

⋄ **Zero-order** methods is one of the key and oldest areas of optimization. There are various zero-order approaches, here we can briefly highlight, e.g., one-dimensional methods (Kiefer, 1953; Nocedal & Wright, 2006) or their high-dimensional analogues (Newman, 1965), ellipsoid algorithms (Yudin & Nemirovskii, 1976) and searches along random directions (Bergou et al., 2020). Currently, the most popular and most studied mechanism behind ZOO methods is the finite-difference approximation of the gradient described in (Polyak, 1987; Flaxman et al., 2005; Nesterov & Spokoiny, 2017). The idea is simple: querying two suf-

---

*Equal contribution  [1]IC, EPFL, Lausanne, Switzerland [2]Optimal Lab, Department of Mechanical and Industrial Engineering, University of Toronto, Toronto, Canada [3]AI Institute, Innopolis University, Innopolis, Russia [4]Basic Research of Artificial Intelligence Laboratory (BRAIn Lab), Moscow, Russia [5]Moscow Independent Research Institute of Artificial Intelligence (MIRAI), Moscow, Russia. Correspondence to: Semyon Chebykin <s.chebykin@mail.utoronto.ca>.

*Proceedings of the 43rd International Conference on Machine Learning*, Seoul, South Korea. PMLR 306, 2026. Copyright 2026 by the author(s).

ficiently close points is essentially equivalent to finding a value of the directional derivative of the function:

$$\langle \nabla f(x), e \rangle \approx \frac{f(x + te) - f(x - te)}{2t}, \qquad (1)$$

where $e$ is a random direction. It can be a random coordinate, a vector from the Euclidean sphere or a sample of the Gaussian distribution. The approximation (1) in turn leads back to the gradient methods or coordinate algorithms of Nesterov (2012). There are, however, several differences:

• First, to get full gradient information, the algorithm would need $d$ queries instead of one gradient oracle call (here $d$ is the dimension of $x$).

• Second, if the ZO oracle is inexact, i.e. only noisy values of function are available, then finite difference schemes can fail if noise components do not cancel out.

The setting of the second point, when function evaluations experience zero-mean additive perturbations, is called *Stochastic ZOO*. The stochasticity, as noted before, is abundant in the modern optimization world. To tackle this issue, additional assumptions about the noise structure are required. Here we briefly discuss two main ideas adopted in the literature, and refer the reader to Section 2 for precise definitions.

In the case of *two-point feedback*, we assume that for a fixed value of the noise variable one can call the stochastic zero-order oracle at least twice. It means that we can compute the finite difference approximation of the following form:

$$\langle \nabla_x f(x, \xi), e \rangle \approx \frac{f(x + te, \xi) - f(x - te, \xi)}{2t} \qquad (2)$$

Such approximation produces an estimate for the directional derivative of a noisy realization $f(\cdot, \xi)$ of the function $f$. As mentioned before, the approximation (2) can be used instead of the (stochastic) gradient in first-order methods. In the case of independent randomness, a large number of works are based on this idea. There are results for both non-smooth and smooth convex problems built on classical and accelerated gradient methods of Nesterov & Spokoiny (2017). In the scope of our paper, we are interested in the results for smooth strongly convex problems from (Dvurechensky et al., 2021), namely estimates on zero-order oracle calls to achieve $\varepsilon$-solution in terms of $\|x - x^*\|$: $\mathcal{O}(\frac{d\sigma_2^2}{\mu^2\varepsilon})$. Here $\sigma_2$ is introduced as the variance of the gradient, i.e. it is assumed that $\mathbb{E}_\xi \nabla f(x, \xi) = \nabla f(x)$ and $\mathbb{E}_\xi \|\nabla f(x, \xi) - \nabla f(x)\|^2 \leq \sigma_2^2$. The main limitation of two-point approach is that several evaluations with the same noise variable are required, which is well suited for problems like empirical risk optimization (Liu et al., 2018), but can be a major barrier for RL or online optimization.

In the *one-point feedback* setting, a more general stochasticity is assumed. In this case, each call to the zero-order oracle generates a new randomness. Now the approximation (1) looks as follows

$$p(x, \xi^{\pm}, e) = \frac{f(x + te, \xi^+) - f(x - te, \xi^-)}{2t} \qquad (3)$$

Using different $\xi^+$ and $\xi^-$ in (3) renders any conditions on the properties of $\nabla f(\cdot, \xi)$ useless. Instead, it is assumed that $\mathbb{E}_\xi f(x, \xi) = f(x)$ and $\mathbb{E}_\xi |f(x, \xi) - f(x)|^2 \leq \sigma_1^2$. With one-point feedback, the major problem is choosing the right shift $t$ for the finite difference scheme. Picking it too small results in an amplification of the additive noise, and taking $t$ too big leads to a poor gradient estimate. Because of this variance trade-off, the optimal rate for methods with one-point approximation is worse than for two-point feedback. In particular, for smooth strongly convex problems we have the following estimate on zero-order oracle calls (Gasnikov et al., 2017): $\mathcal{O}(\frac{d^2\sigma_1^2}{\mu^3\varepsilon^2})$.

Although zero-order gradient approximation schemes suffer from high variance, there is a surprising property that makes them superior in *non-smooth* optimization (Gasnikov et al., 2022; Qiu et al., 2023; Shamir, 2017). The idea goes back to the 70s and utilizes the fact that

$$\mathbb{E}[e \cdot p(x, \xi^{(\pm)}, e)] = \frac{1}{d} \nabla f_t(x),$$

where $f_t$ is a *smoothed* function, defined as

$$f_t(x) = \mathbb{E}_r [f(x + tr)] \text{ with } r \sim RB_2^d.$$

In fact, it can be shown that $f_t$ is $\frac{\sqrt{d}G}{t}$-smooth if $f$ is $G$-Lipschitz. This makes zero-order approximation a suitable candidate for a stochastic gradient of $f_t$. Optimizing this function with a first-order method produces some solution, but it may not be the optima of $f$ (Gasnikov et al., 2022). From this point, there is a game – for small $t$ the functions $f$ and $f_t$ are closer and for big $t$ the function $f_t$ is easier to optimize as it gets smoother.

In more recent works, there have been many improvements in theoretical understanding of ZO methods. The authors consider higher-order smoothness of the underlying function (Akhavan et al., 2024), tackle non-convex non-smooth problems (Qiu et al., 2023), take arbitrary Bregman geometry to benefit in terms of oracle complexity (Shamir, 2017; Gorbunov et al., 2022), and come up with sharp information-theoretic lower bounds to understand computational limits (Duchi et al., 2015; Akhavan et al., 2020). But none of them consider Markovian stochasticity.

⋄ **Markovian first-order methods.** While the literature on stochastic optimization with i.i.d. noise is extensive, research addressing the Markovian setting remains relatively sparse. In our paper, we focus on the most "friendly" type of uniformly geometrically ergodic Markov chains (see Section 2 for precise definitions).

Duchi et al. (2012) conducted pioneering work on non-i.i.d. noise, investigating the Ergodic Mirror Descent algorithm and establishing optimal convergence rates for non-smooth convex problems. For smooth problems there were different attempts to get record-breaking estimates on the first-order oracle (Doan et al., 2020; Doan, 2022; Zhao, 2023; Even, 2023). Finally, the optimal results were obtained for both convex and non-convex problems in the works of Beznosikov et al. (2024); Solodkin et al. (2024). In particular, for smooth strongly convex objectives under Markovian noise the authors give the complexity of the form: $\mathcal{O}\left(\frac{\tau\sigma_2^2}{\mu^2\varepsilon}\right)$, where $\tau$ is defined as the mixing time of the corresponding Markov chain (see Section 2). Note that these works utilize Multilevel Monte Carlo (MLMC) batching technique, which helps to effectively interact with Markovian noise. We will need this approach as well. Note that it was first considered in Markovian gradient optimization by Dorfman & Levy (2022) for automatic adaptation to unknown $\tau$.

$\diamond$ **Hypothesis.** The complexity estimate for strongly convex first-order stochastic methods is $\mathcal{O}\left(\frac{\sigma_2^2}{\mu^2\varepsilon}\right)$ (Moulines & Bach, 2011; Needell et al., 2014). Lower bounds for the same class of problems and methods show that the result is unimprovable (Yudin & Nemirovskii, 1976). As mentioned before, the transition from i.i.d. stochasticity to Markovian stochasticity increases the estimate by $\tau$ times. This result is also optimal as shown by Beznosikov et al. (2024). At the same time, going from gradient oracle to zero-order methods adds a multiplier $d$ in the two-point feedback and $d^2/\varepsilon$ in the one-point case. And this estimate is unimprovable as well (Akhavan et al., 2020; Duchi et al., 2015).

The hypothesis arises that the transition to zero-order Markov optimization adds two multipliers at once: $d\tau$ and $d^2\tau/\varepsilon$ for two- and one-point, resulting in $\mathcal{O}\left(\frac{d\tau\sigma_2^2}{\mu^2\varepsilon}\right)$ and $\mathcal{O}\left(\frac{d^2\tau\sigma_1^2}{\mu^2\varepsilon^2}\right)$ complexities respectively. It is illustrated in the following diagram for two-point feedback:

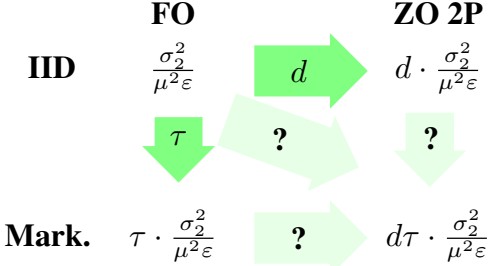

### 1.2. Our contribution

Our main contribution is the answer to the hypothesis above: *surprisingly, it is not true*. In more detail:

$\diamond$ **Accelerated SGD.** We present the first analysis of Zero-Order Accelerated SGD under Markovian noise, considering both two-point and one-point feedback. Contrary to the expected multiplicative scaling of convergence rates with both dimensionality and mixing time, our analysis reveals a significant acceleration, as presented in Table 1. It turns out that if $\tau$ is smaller than $d$, our results do not differ at all from the gradient-free methods with independent stochasticity. The key technique behind this acceleration is described in Section 3.1. The theory is also numerically validated in Section 4.

Table 1. Summary of upper bounds. For notation, see Table 2.

| | **Smooth** | | **Non-smooth** | |
| | IID | Markovian | IID | Markovian |
|---|---|---|---|---|
| **FO** | $\frac{\sigma_2^2}{\mu^2\varepsilon}$ [1] | $\tau\frac{\sigma_2^2}{\mu^2\varepsilon}$ [2] | $\frac{G^2}{\mu^2\varepsilon}$ [3] | $\tau\frac{G^2}{\mu^2\varepsilon}$ [4] |
| **ZO 2P** | $d\frac{\sigma_2^2}{\mu^2\varepsilon}$ [5] | $(d+\tau)\frac{\sigma_2^2}{\mu^2\varepsilon}$ | $d\frac{G^2}{\mu^2\varepsilon}$ [6] | $(d+\tau)\frac{G^2}{\mu^2\varepsilon}$ |
| **ZO 1P** | $d^2\frac{L\sigma_1^2}{\mu^3\varepsilon^2}$ [7] | $d(d+\tau)\frac{L\sigma_1^2}{\mu^3\varepsilon^2}$ | $d^2\frac{\sigma_1^2 G^2}{\mu^4\varepsilon^3}$ [8] | $d(d+\tau)\frac{\sigma_1^2 G^2}{\mu^4\varepsilon^3}$ |

[1]: (Robbins & Monro, 1951); [2]: (Beznosikov et al., 2024); [3]: (Shamir & Zhang, 2013); [4]: (Duchi et al., 2012)[1]; [5]: (Hazan & Kale, 2014); [6]: (Gasnikov et al., 2022); [7]: (Akhavan et al., 2020)[2]; [8]: (Gasnikov et al., 2017); [Rest]: (This paper);

$\diamond$ **Non-smooth problems.** We also consider non-smooth problems with Markovian noise. Using the smoothing technique we come up with a corresponding upper bounds in this case, as shown in Table 1. The details of these bounds are presented in Section B.2.

$\diamond$ **Computational efficiency.** First, as noted above, our method gives the same oracle complexity for any $\tau \leq d$. Moreover, if we assume that calling a zero-order oracle is $d$ times cheaper than computing the corresponding gradient, then the gradient method with Markov noise will require resources proportionally to $d\cdot\tau$ — the cost of one oracle call is $d$ and the complexity scales as $\tau$ for the first-order method from Table 1. At the same time, the resource complexity of our zero-order method is proportional to $d+\tau$.

$\diamond$ **Lower bounds.** In Section 3.3 we establish the first information-theoretic lower bounds for solving Markovian optimization problems with one-point and two-point feedback. Our results match the convergence guarantee of our algorithm up to logarithmic factors, showing that the analysis is accurate and no further improvement is possible.

## 2. Preliminaries

We are now ready for a more formal presentation. In this paper, we study the minimization problem

$$\min_{x\in\mathbb{R}^d} f(x) := \mathbb{E}_{Z\sim\pi}\left[F(x,Z)\right], \qquad (4)$$

---

[1]Authors considered general convex case. Using standard restart technique, we get the corresponding bound in the strongly convex case.

[2]The noise is assumed to be point-independent.

[3]By construction, for any $A \in \mathcal{Z}$, we have $\mathbb{P}_\xi(Z_k \in A \mid Z_{k-1}) = \mathrm{Q}(Z_{k-1}, A), \quad \mathbb{P}_\xi$-a.s.

*Table 2.* Notations & Definitions

| Sym. | Definition | Sym. | Definition |
|---|---|---|---|
| $\|\cdot\|, \langle\cdot,\cdot\rangle$ | Norm, dot product, assumed Euclidean by default | $\varepsilon$ | $\|x - x^*\|^2$ |
| $\mathsf{Z}, \mathcal{Z}$ | Complete separable metric space, its Borel $\sigma$-algebra | $d$ | Problem dimension |
| $\mathsf{Q}$ | Markov kernel on $\mathsf{Z} \times \mathcal{Z}$ | $L$ | Gradient's Lipshitz constant |
| $\mathbb{P}_\xi, \mathbb{E}_\xi$ | Probability, Expectation under initial distribution $\xi^3$ | $\mu$ | Strong convexity constant |
| $\{Z_k\}$ | Canonical process with kernel Q | $G$ | Function's Lipshitz constant |
| $RB_2^d, RS_2^d$ | Uniform distribution on unit a $\ell_2$-ball, -sphere | $\sigma_1^2$ | $\|F(x, Z) - f(x)\|^2 \le \sigma_1^2$ |
| $e$ | Random direction, $e \sim RS_2^d$ | $\sigma_2^2$ | $\|\nabla F(x, Z) - \nabla f(x)\|^2 \le \sigma_2^2$ |
| $a_n \lesssim b_n$ | $\exists c \in \mathbb{R}$ (universal constant): $a_n \le cb_n$ for all $n$ | $\tau$ | Mixing time of $Z$ |
| $a_n \simeq b_n$ | $a_n \lesssim b_n$ and $b_n \lesssim a_n$ | $g, \hat{g}$ | Gradient estimators |
| $T = \tilde{\mathcal{O}}(S)$ | $T \le poly(\log S) \cdot S$ as $\varepsilon \to 0$ | $f_t(x)$ | $\mathbb{E}_r\left[f(x + tr)\right], r \sim RB_2^d$ |

where $\pi$ is an unknown distribution (see Assumption 3) and access to the function $f$ (not to its gradient $\nabla f$) is available through a stochastic one-point or two-point oracle $F(x, Z)$.

In our analysis, we will use a set of assumptions on the underlying function $f$ and its oracle, starting with smoothness and convexity:

**Assumption 1.** *The function $f$ is $L$-smooth on $\mathbb{R}^d$ with $L > 0$, i.e., it is differentiable and there is a constant $L > 0$ such that the following inequality holds for all $x, y \in \mathbb{R}^d$:*

$$\|\nabla f(x) - \nabla f(y)\| \le L\|x - y\|.$$

In the two-point feedback setting, we require the following generalization:

**Assumption 1′.** *For all $Z \in \mathsf{Z}$ the function $F(\cdot, Z)$ is $L$-smooth on $\mathbb{R}^d$.*

Note that the uniform 1′ implies 1.

**Assumption 2.** *The function $f$ is continuously differentiable and $\mu$-strongly convex on $\mathbb{R}^d$, i.e., there is a constant $\mu > 0$ such that the following inequality holds for all $x, y \in \mathbb{R}^d$:*

$$\frac{\mu}{2}\|x - y\|^2 \le f(x) - f(y) - \langle \nabla f(y), x - y \rangle. \quad (5)$$

We now turn to assumptions on the sequence of noise states $\{Z_i\}_{i=0}^\infty$. Specifically, we consider the case where $\{Z_i\}_{i=0}^\infty$ forms a time-homogeneous Markov chain. Let Q denote the corresponding Markov kernel. We impose the following assumption on Q to characterize its mixing properties:

**Assumption 3.** *$\{Z_i\}_{i=0}^\infty$ is a stationary Markov chain on $(\mathsf{Z}, \mathcal{Z})$ with Markov kernel Q and unique invariant distribution $\pi$. Moreover, Q is uniformly geometrically ergodic with mixing time $\tau \in \mathbb{N}$, i.e., for every $k \in \mathbb{N}$, total variation after $k$ steps decays as*

$$\sup_{z, z' \in \mathsf{Z}} (1/2)\|\mathsf{Q}^k(z, \cdot) - \mathsf{Q}^k(z', \cdot)\|_{\mathsf{TV}} \le (1/4)^{\lfloor k/\tau \rfloor}. \quad (6)$$

Assumption 3 is common in the literature on Markovian stochasticity (Duchi et al., 2012; Doan et al., 2020; Dorfman & Levy, 2022; Beznosikov et al., 2024; Solodkin et al.,

2024). It includes, for instance, irreducible aperiodic finite Markov chains (Even, 2023). The mixing time $\tau$ reflects how quickly the distribution of the chain approaches stationarity, providing a natural measure of the temporal dependence in the data.

Next, we specify our assumptions on the oracle. As discussed in Section 1.1, these assumptions differ based on the type of feedback.

**Assumption 4** (for one-point)**.** *For all $x \in \mathbb{R}^d$ it holds that $\mathbb{E}_\pi[F(x, Z)] = f(x)$. Moreover, for all $Z \in \mathsf{Z}$ and $x \in \mathbb{R}^d$ it holds that*

$$|F(x, Z) - f(x)|^2 \le \sigma_1^2,$$

**Assumption 4′** (for two-point)**.** *For all $x \in \mathbb{R}^d$ it holds that $\mathbb{E}_\pi[\nabla F(x, Z)] = \nabla f(x)$. Moreover, for all $Z \in \mathsf{Z}$ and $x \in \mathbb{R}^d$ it holds that*

$$\|\nabla F(x, Z) - \nabla f(x)\|^2 \le \sigma_2^2.$$

Recent works on stochastic ZOO methods have considered milder assumptions, such as bounded variance (see Section 1.1). However, the uniform boundedness assumed in Assumptions 4 and 4′, is standard in analyses under Markovian noise (Duchi et al., 2012; Doan et al., 2020; Dorfman & Levy, 2022; Beznosikov et al., 2024; Solodkin et al., 2024). These assumptions can be relaxed under stronger conditions, e.g., uniform convexity and smoothness of $F(\cdot, Z)$ (Even, 2023).

Assumptions 3 and 4 allow us to reduce the variance of the noise via batching, similarly the to i.i.d. setting. This is captured in the following technical lemma:

**Lemma 1.** *Let Assumptions 3 and 4(4′) hold. Then for any $n \ge 1$ and $x \in \mathbb{R}^d$ and any initial distribution $\xi$ on $(\mathsf{Z}, \mathcal{Z})$, we have*

$$\mathbb{E}_\xi\left[\frac{1}{n}\sum_{i=1}^n F(x, Z_i) - f(x)\right]^2 \lesssim \frac{\tau}{n}\sigma_1^2 \quad and$$

$$\mathbb{E}_\xi\left\|\frac{1}{n}\sum_{i=1}^n \nabla F(x, Z_i) - \nabla f(x)\right\|^2 \lesssim \frac{\tau}{n}\sigma_2^2.$$

## 3. Main results

### 3.1. Batching technique

In this section, we describe the main tools used to establish the $(d + \tau)$-type scaling of the error rate. We will focus on reducing the variance and bias of gradient estimators using a specialized batching approach.

We begin by fixing a common building block of our gradient estimators at a point $x$ for both one-point and two-point feedback, as introduced in Section 1.1:

$$\hat{g}(x, Z^{(\pm)}, e) = d \cdot p(x, Z^{(\pm)}, e) \cdot e =$$

$$= e \cdot \begin{cases} d\dfrac{F(x + te, Z^+) - F(x - te, Z^-)}{2t} & \text{(one-point)}, \\[2mm] d\dfrac{F(x + te, Z) - F(x - te, Z)}{2t} & \text{(two-point)}. \end{cases}$$

These estimators exhibit a twofold randomness that affects how rapidly they concentrate around the true gradient, as we will discuss below.

For clarity, we focus our discussion on the one-point case, although our conclusions extend to the two-point case as well.

A widely used variance reduction technique is *mini-batching*, where one computes $F(x, Z_i)$ over a batch of noise variables $\{Z_i\}_{i=1}^n$. The mini-batch gradient estimator is given by:

$$\hat{g}_{mb}(x) = \frac{1}{n}\sum_{i=1}^{n} \hat{g}(x, Z_i^{\pm}, e) = d \overbrace{\left(\frac{1}{n}\sum_{i=1}^{n} p(x, Z_i^{\pm}, e)\right)}^{p_{mb}} \cdot e.$$

Let us estimate the scaling of its variance $\mathbb{E}_e \mathbb{E}_Z \|\hat{g}_{mb} - \nabla f\|^2$ with the noise level $\sigma_1^2$. As $\mathbb{E}_Z \hat{p}_{mb} \approx \frac{f(x+te) - f(x-te)}{2t} \approx \langle \nabla f, e \rangle$ we would like to estimate the following for any fixed direction $e$:

$$\mathbb{E}_Z\left[p_{mb}(x) - \langle \nabla f, e \rangle\right]^2 \approx \tag{7}$$

$$\frac{1}{t^2}\mathbb{E}_Z\left[\frac{1}{n}\sum_{i=1}^{n} F(x + te, Z_i^+) - f(x + te)\right]^2 \overset{(1)}{\approx} \frac{\tau}{n}\frac{\sigma_1^2}{t^2}.$$

The second transition assumes the "worst-case" scenario, where Lemma 1 is tight. For lazy Markov chains (discussed below), the inequality indeed turns into an equality. With that, we bound the variance:

$$\mathbb{E}_e \mathbb{E}_Z \|\hat{g}_{mb} - \nabla f\|^2 \geq \mathbb{E}_e \mathbb{E}_Z \|\hat{g}_{mb} - \mathbb{E}_Z \hat{g}_{mb}\|^2 =$$

$$\mathbb{E}_e \mathbb{E}_Z \|d \cdot [p_{mb} - \mathbb{E}_Z[p_{mb}]] \cdot e\|^2 = \tag{8}$$

$$d^2 \mathbb{E}_e \mathbb{E}_Z |p_{mb} - \langle \nabla f, e \rangle|^2 \overset{(7)}{\approx} \frac{d^2 \tau \sigma_1^2}{nt^2}.$$

**Can the mini-batching scheme be improved?**

This subsection explores an unexpected source of improvement that contradicts our initial hypothesis. Specifically, we identify an inefficiency in the current use of samples $Z_i$, which becomes evident from two perspectives. Equation (8) shows the variance scales as $\frac{\tau}{n}$. If we could reduce $\tau$ by a factor of $k$, we would need $k$-times fewer samples to maintain the same variance. This leads us to the idea of sparsified sampling. We partition the Markov noise chain $\{Z_i\}$ into $k$ subchains $\{Z_{k \cdot i + r}\}$ for $r = 0 \ldots k - 1$. This corresponds to a mixing time of $\lceil \frac{\tau}{k} \rceil$ for each subchain (see (3)), effectively reducing temporal correlation - a natural consequence of sampling every $k$-th element of the original chain. Thus, sampling from any single subchain could yield a $\min(k, \tau)$-fold reduction in the number of samples needed (although such procedure would still require all intermediate oracle calls, yielding no computational speedup).

For a concrete illustration of that inefficiency, consider a lazy Markov chain that remains in the same state for (an average of) $\tau$ steps before transitioning uniformly at random. In such a case, all oracle queries $F(x, Z)$ for a fixed $x$ return the same value for $\tau$ consecutive steps. Therefore, retaining only every $\tau$-th estimate $\hat{g}$ would yield a mini-batch of equivalent quality.

In summary, we observe that the mini-batching scheme could, in principle, operate just as effectively by retaining only every $k$-th sample and discarding the rest. This might suggest that better utilization of the samples is possible. First order methods, nevertheless, are unable to exploit this redundancy (as shown by (Beznosikov et al., 2024)'s lower bound) and are effectively forced to wait out the $\tau$-step mixing window. In contrast, we can exploit this structure by querying finite differences along different directions to estimate the gradient better. Specifically, we construct $d$ subchains, where $r$-th subchain $Z_{d \cdot i + r}$ is used for oracle calls along $r$-th coordinate: $\frac{F(x + te_r, Z) - F(x - te_r, Z)}{2t}$. Thus the full gradient is restored coordinate-wise.

Let us estimate the resulting variance reduction. First, we achieve a $d$-fold reduction by reconstructing all $d$ gradient coordinates. Second, each coordinate now operates on a chain with mixing time $\lceil \frac{\tau}{d} \rceil$, yielding an additional factor of $\min(d, \tau)$. However, because batches are now split across $d$ coordinates, each batch is $d$ times smaller than before, introducing a factor of $d$ loss. The net variance reduction is therefore $\min(d, \tau)$, and the final scaling becomes $d \cdot \frac{d\tau}{\min(d, \tau)} = d \cdot \max(d, \tau) \simeq d(d + \tau)$.

**Random directions**

This insight can be extended to a simpler yet equally effective method. Instead of assigning directions deterministically, we associate each sample with a random direction

$e \in RS_2^d$, forming the estimator:

$$\hat{g}_{rd}[n](x, Z, e) = \frac{1}{n} \sum_{i=1}^{n} \hat{g}(x, Z_i, e_i).$$

While the above discussion was intuitive, we now outline a more formal approach (see Lemma 5 for details). As lazy Markov chain is effectively equivalent to stochastic i.i.d. $\tau$-point feedback setting, we follow Corollary 2 of (Duchi et al., 2015), which decomposes the total variance into two terms:

$$\mathbb{E}\|\hat{g}_{rd} - \nabla f(x)\|^2 \leq 2\mathbb{E}\|\hat{g}_{rd} - v\|^2 + 2\mathbb{E}\|v - \nabla f(x)\|^2$$

with $v = \mathbb{E}_e \hat{g}_{rd}$. Each of the two terms individually eliminates one factor from the $d^2\tau$ dependence.

The first term $\mathbb{E}\|\hat{g}_{rd} - \mathbb{E}_e\hat{g}_{rd}\|^2 =$

$$= \mathbb{E}_Z\mathbb{E}_e \left\| \frac{1}{n} \sum_{i=1}^{n} \underbrace{[\hat{g}(x, Z_i, e_i) - E_{e_i}\hat{g}(x, Z_i, e_i)]}_{\mathbb{E}_e[\cdot]=0, \text{ independent w.r.t. } e} \right\|^2$$

$$= \frac{1}{n^2} \sum_{i=1}^{n} \mathbb{E}\|\hat{g}(x, Z_i, e_i) - \mathbb{E}_{e_i}\hat{g}(x, Z_i, e_i)\|^2$$

is independent of $\tau$ since Assumption 4 bounds each term directly.

For the second term, we observe that $\mathbb{E}_e\hat{g}_{rd} = \mathbb{E}_e\hat{g}_{mb}$, and thus the bound involves $\mathbb{E}\|\mathbb{E}_e\hat{g}_{mb} - \nabla f(x)\|^2$. This is crucially different from the $d^2\tau$ dependence that appeared in the mini-batch case, when we considered $\mathbb{E}\|\hat{g}_{mb} - \nabla f(x)\|^2$. Intuitively, the expectation over directions helps recover the full gradient rather than a directional component, thereby reducing variance with respect to $d$.

**Multilevel Monte Carlo**
The estimator $\hat{g}_{rd}$ is not our final construction. While it controls variance, the temporal correlation in noise may introduce significant bias. A well-established approach to mitigating this is MLMC, widely used in the statistical literature (Glynn & Rhee, 2014; Giles, 2008), and more recently in gradient optimization (Dorfman & Levy, 2022; Beznosikov et al., 2024). Here is our interpretation.

With parameters $J, l, M, B$ from Table 3, $\{Z_i\} - 2^J l$ samples from $Z$ and $\{e_i\}$ – random directions we introduce MLMC estimator:

$$\hat{g}_{ml}(x) = \hat{g}_{rd}[2^0 l](x) +$$

$$+ \begin{cases} 2^J \left[ \hat{g}_{rd}[2^J l](x) - \hat{g}_{rd}[2^{J-1} l](x) \right], & \text{if } 2^J \leq M \\ 0, & \text{otherwise.} \end{cases}$$

To ease understanding of the formula above consider an example with $l = 1, M = \infty, B = 1$. Enumerate

base estimates as $g_1, g_2, \ldots$, then MLMC estimate will be $\hat{g}_{ml} = g_1 + 2(\frac{g_1+g_2}{2} - g_1) = g_2$ with prob. $1/2$; $g_1 + (g_3 + g_4 - g_1 - g_2)$ with prob. $1/4$ and so on. Parameter $M$ is the upper bound on the number of base estimates used. Parameter $l$ transforms the base estimator into a sequence of $l$ base estimators, effectively stretching everything $l$ times. Finally, $B$ serves as a hyperparameter that can multiplicatively increase $l$. $\hat{g}_{ml}$ is our final gradient estimator, with the following guarantees:

**Lemma 2** (for one-point). *Let Assumptions 1, 3 and 4 hold. For any initial distribution[1] $\xi$ on $(\mathsf{Z}, \mathcal{Z})$ the gradient estimates $\hat{g}_{ml}$ satisfy $\mathbb{E}[\hat{g}_{ml}] = \mathbb{E}\left[ \hat{g}_{rd}\left[ 2^{\lfloor \log_2 M \rfloor} l \right] \right]$. Moreover,*

$$\|\nabla f_t(x) - \mathbb{E}[\hat{g}_{ml}(x)]\|^2 \lesssim \frac{d\tau\sigma_1^2}{t^2 BM},$$

$$\mathbb{E}\|\nabla f_t(x) - \hat{g}_{ml}(x)\|^2 \lesssim \frac{d\|\nabla f(x)\|^2 + d^2 L^2 t^2 + \frac{d(d+\tau)\sigma_1^2}{t^2}}{B}$$

One can note that although $\hat{g}_{ml}$ requires on average $l \log_2 M \simeq \log_2^2 M \cdot B$ oracle calls, the variance is only reduced by a factor of $B$. In contrast, the bias is reduced significantly – by a factor of $BM$.

### 3.2. Algorithm

We now present the full version of Algorithm 1, which incorporates the gradient estimators discussed in the previous section and uses a slightly modified variant of Nesterov's Accelerated Gradient Descent at its core.

---
**Algorithm 1** `Randomized Accelerated ZO GD`

---
1: **Initialization:** $x_f^0 = x^0$; see Table 3.
2: **for** $k = 0, 1, 2, \ldots, N - 1$ **do**
3:   $\quad x_g^k = \theta x_f^k + (1 - \theta)x^k$
4:   $\quad$ Sample $J, \{e_i\}, \left\{ F(x_g^k \pm te_i, Z_i^{(\pm)}) \right\}$
5:   $\quad$ Calculate $\hat{g}^k = \hat{g}_{ml}(x_g^k)$
6:   $\quad x_f^{k+1} = x_g^k - p\gamma\hat{g}^k$
7:   $\quad x^{k+1} = \eta x_f^{k+1} + (p - \eta)x_f^k +$
         $\quad\quad\quad + (1 - p)(1 - \beta)x^k + (1 - p)\beta x_g^k$
8: **end for**

---

While technically we prove four separate upper bounds covering both one- and two-point feedback under smooth and non-smooth assumptions, they follow the same scheme which we will illustrate in the one-point smooth case.

Lemma 4 establishes key properties of the smoothed objective function. Lemma 5 provides bounds on the bias and

---

[1]Note that $\hat{g}_{ml}$ (specifically $Z_1$) indirectly depends on the chain's initial distribution. As our algorithm is going to repeatedly call $\hat{g}_{ml}$, next iteration's initial distribution is current iteration's final distribution. This fact makes the estimates correlated. We sidestep this problem by assuming any initial distribution.

[1]in smooth and non-smooth settings respectively

*Table 3.* Parameters of Algorithm 1

| Hyperparameters | | Momentums | | Batch hidden parameters | |
|---|---|---|---|---|---|
| $\gamma$ | Stepsize, $\in (0; \frac{3}{4L}]$ | $\beta$ | $\sqrt{\frac{4p^2\mu\gamma}{3}}$ | $2^J l$ | Batch size. If $2^J > M$, then 0 |
| $t$ | Approximation step | $\eta$ | $\sqrt{\frac{3}{\mu\gamma}}$ | $J$ | Random, $J \sim \text{Geom}(1/2)$ |
| $B$ | Batch size multiplier | $\theta$ | $\frac{p\eta^{-1}-1}{\beta p\eta^{-1}-1}$ | $M$ | Batch size limit, $M = \frac{1}{p} + \frac{2}{\beta}$ |
| $N$ | Number of iterations | $p$ | $\frac{B}{B+d}$ or[1] const | $l$ | $(\lfloor \log_2 M \rfloor + 1) \cdot B$ |

variance of the baseline estimator $\hat{g}_{rd}$. Lemma 2 then quantifies how the MLMC scheme amplifies or reduces these statistics. Finally, in Section D.4, we combine the results of these lemmas to prove the first part of Theorem 1, bounding Algorithm 1's error. By tuning the parameters appropriately, we obtain the following iteration complexity bound:

**Theorem 1.** *Let Assumptions 1 to 4 hold, and consider problem* (4) *solved by Algorithm 1. Then, for any target accuracy $\varepsilon$ and batch size multiplier $B$ (see Tables 2 and 3 for notation), and for a suitable choice of $\gamma, t, p$, the expected[2] number of one-point oracle calls required to ensure $\mathbb{E}\|x^N - x^*\|^2 \leq \varepsilon$ is bounded by*

$$B \cdot \tilde{\mathcal{O}}\left(\max\left[1, \frac{d}{B}\right]\sqrt{\frac{L}{\mu}}\log\frac{1}{\varepsilon} + \frac{Ld(d+\tau)\sigma_1^2}{B\mu^3\varepsilon^2}\right) .$$

**Theorem 1'.** *Let Assumptions 1' to 4' hold, and consider problem* (4) *solved by Algorithm 1. Then, for any target accuracy $\varepsilon$ and batch size multiplier $B$ (see Tables 2 and 3 for notation), and for a suitable choice of $\gamma, t, p$, the expected number of two-point oracle calls required to ensure $\mathbb{E}\|x^N - x^*\|^2 \leq \varepsilon$ is bounded by*

$$B \cdot \tilde{\mathcal{O}}\left(\max\left[1, \frac{d}{B}\right]\sqrt{\frac{L}{\mu}}\log\frac{1}{\varepsilon} + \frac{(d+\tau)\sigma_2^2}{B\mu^2\varepsilon}\right) .$$

*Remark.* The *iteration complexity* of the algorithm, i.e., the number of iterates $x^k$ generated (equal to the oracle complexity divided by $B$), is bound by $\tilde{\mathcal{O}}\left(\sqrt{\frac{L}{\mu}}\log\frac{1}{\varepsilon}\right)$ as the batch size multiplier $B$ goes to infinity. This matches the optimal convergence rates for optimization with *exact* gradients (Nesterov, 1983).

### 3.3. Lower bounds

Here we present theorems demonstrating that no algorithm can asymptotically outperform Algorithm 1 in the smooth, strongly convex setting with either one- or two-point feedback.

**Theorem 2.** *(Lower bounds) For any (possibly randomized) algorithm that solves the problem* (4)*, there exists a function $f$ that satisfies Assumptions 1 to 4 (1' to 4'), s.t. in order to achieve $\varepsilon$-approximate solution in expectation $\mathbb{E}\|x^N -$*

$x^*\|^2 \leq \varepsilon$ *with one- or two-point oracle, the algorithm respectively needs at least*

$$\Omega\left(\frac{d(d+\tau)\sigma_1^2}{\mu^2\varepsilon^2}\right) \quad or \quad \Omega\left(\frac{(d+\tau)\sigma_2^2}{\mu^2\varepsilon}\right) \quad oracle\ calls.$$

*Remark.* These results assume bounded second moments rather than uniform noise bounds. We explain how to adapt them to our setting, incurring only logarithmic overheads, in Section F.2.

**Discussion.** We now compare our results to existing work. Akhavan et al. (2024) analyze a special case of the one-point setting where the noise is independent of the query points. This aligns with our one-point oracle model and allows i.i.d. sampling as a Markov chain with fixed mixing time $\tau = 1$. The only factor they do not consider is $\sigma_1^2$, which, however, appears in their proof with additional $\mu^2$ factor if used with scaled Gaussian noise. We discuss this further in Appendix F.

In the work of Beznosikov et al. (2024), a first-order Markovian oracle is considered, but the hard instance problem is a one-dimensional quadratic function, which makes first-order and zero-order information equivalent. Their result therefore corresponds to the $d = 1$ case in the two-point regime. Duchi et al. (2015) provide tight lower bounds for general convex functions under two-point feedback. Their techniques can be extended to the strongly convex case by incorporating a shared quadratic component across the hard instances, as detailed in Appendix F, Theorem 10, yielding the bound we state for the two-point oracle with $\tau = 1$.

Our novel contribution lies in establishing a lower bound that scales as $d\tau$ in the one-point regime for large $\tau$; see Theorem 8. While our analysis relies on classical tools such as multidimensional hypothesis testing, the Markovian structure requires new bound on distances between joint distributions and the use of clipping. Detailed proofs, discussions, and further remarks on clipping appear in Appendix F.

## 4. Experiments

This section empirically supports our theoretical convergence rates and lower bounds, with particular focus on the stochastic component where we claim linear scaling in $d+\tau$

---

[2]for high-probability bound, see Section B.3

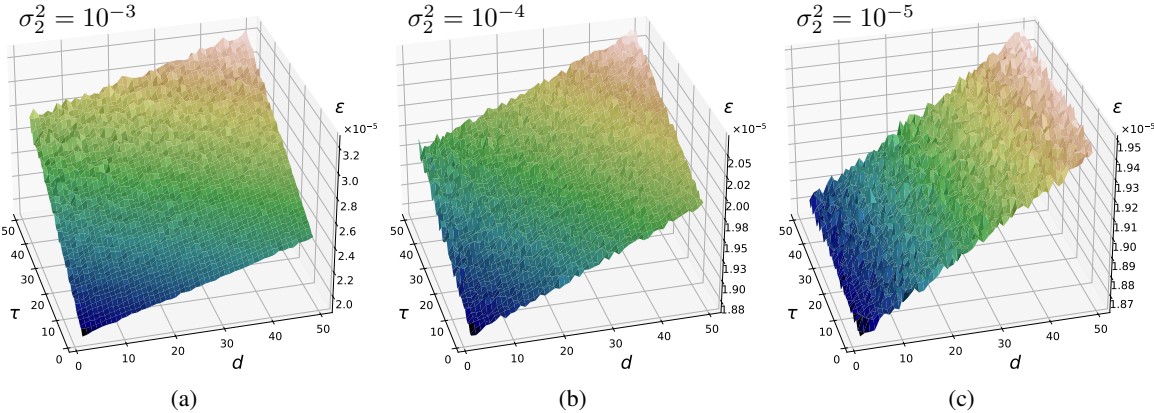

*Figure 1.* Optimization error $\varepsilon = \|x^N - x^*\|^2$ after $N = 10^3$ iterations. Starting point error $\|x_0 - x^*\|^2 = 10^{-2}$. Stepsize $\gamma = 10^{-3}$, $t = 10^{-5}$. The results are averaged over $10^4$ runs.

instead of $d\tau$.

**Setup.** Our setup repeats the problem we used to prove the lower bounds (see Appendix F and (Shapiro et al., 2009)). We consider a quadratic objective $f(x) = \frac{1}{2}\|x\|^2$ and a two-point Markovian oracle $F(x, Z) = f(x) + \langle x, Z \rangle$. The noise sequence $\{Z_i\}$ is a lazily updated standard Gaussian vector with variance $\sigma_2^2$. Figure 1 illustrates how the optimization error of Algorithm 1 scales with mixing time, problem dimension, and different values of $\sigma_2^2$.

**Discussion.** The results confirm the linear dependence of the error on both the problem dimension $d$ and the mixing time $\tau$. The noise parameter $\sigma^2$ controls the influence of the stochastic part. In Fig. (a), where $\sigma_2^2 = 10^{-3}$, the stochastic component dominates, while in Fig. (c), with $\sigma_2^2 = 10^{-5}$, it is negligible. Fig. (b) shows an intermediate regime that smoothly interpolates between the two, yet maintains the linear scaling. The deterministic part (c) shows no dependence on mixing time, but grows linearly with $d$, which aligns with our theory (Theorem $1'$). The stochastic part (a) scales as $(d + \tau)$, also matching the bound from the Theorem $1'$.

## 5. Conclusion

We study derivative-free stochastic optimization with Markovian noise sequence in one- and two- point settings. We propose a gradient estimation scheme with reduced variance that allows us to achieve additive, not multiplicative complexity w.r.t. problem dimension and mixing time. We establish the optimality of the claimed convergence rate by proving corresponding lower bounds, and validate the theory numerically. Future work might apply our technique to problems with different (non-convex, PL) or more relaxed (ergodicity, bounded variance) assumptions. Finally, a more practically oriented algorithm, which does not require knowledge of mixing time in advance is of interest.

## Impact Statement

This paper presents work whose goal is to advance the field of Machine Learning. There are many potential societal consequences of our work, none which we feel must be specifically highlighted here.

## Acknowledgements

The work of Aleksandr Beznosikov was supported by the Ministry of Economic Development of the Russian Federation (agreement No. 139-15-2025-013, dated June 20, 2025, IGK 000000C313925P4B0002).

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

## A. Appendix overview

In this section, the overall structure of the technical appendices is presented.

In Appendix B, we introduce the additional adversarial robustness of the Algorithm 1 and present a formal statement of our results in the non-smooth case.

In Appendix C, we define the shorthanded notation used in the proof of upper bounds.

In Appendices D and E, we gradually introduce all lemmas and proofs of our theorems in one-point and two-point setting respectively, for both smooth and non-smooth problems.

In Appendix F we present our lower bounds and provide a more detailed overview of the related results.

Finally, in Appendix G, we formally state the common-knowledge facts that we use.

## B. Additional results

### B.1. Adversarial noise

In addition to the main results that show optimal scaling with the stochastic noise, we also prove a *robustness* of our algorithm. Precisely, the oracle $F$ considered in this paper may return its values with an additive, non-random, potentially adversarial error $\Delta(x) \leq \Delta$.

$$\hat{F}(x, Z) = F(x, Z) + \Delta(x). \tag{9}$$

We will prove that this have no effect of the convergence guarantees of our algorithm for any $\Delta$ within a tolerable threshold. This threshold varies between smooth and non-smooth case, but not between one-point and two-point settings. The precise bounds for $\Delta$ are presented in the theorems in Appendices D and E.

### B.2. Non-smooth

In the non-smooth case, we consider a similar set of assumptions, however $f$ is no longer necessarily smooth or even differentiable.

**Assumption 5.** *The function $f$ is $\mu$-strongly convex on $\mathbb{R}^d$, i.e., there is a constant $\mu > 0$ such that the following inequality holds for all $x, y \in \mathbb{R}^d$ and $\lambda \in [0; 1]$:*

$$f(\lambda x + (1 - \lambda)y) \leq \lambda f(x) + (1 - \lambda)f(y) - \lambda(1 - \lambda)\frac{\mu}{2}\|x - y\|^2$$

**Assumption 6.** *The function $f$ is $G$-Lipschitz on $\mathbb{R}^d$, i.e., there is a constant $G > 0$ such that the following inequality holds for all $x, y \in \mathbb{R}^d$:*

$$|f(x) - f(y)| \leq G\|x - y\|.$$

Again, for the two-point case, we need the generalization:

**Assumption 6′.** *For all $Z \in \mathsf{Z}$ the function $F(\cdot, Z)$ is $G$-Lipschitz on $\mathbb{R}^d$.*

Regarding the noise levels, we keep Assumption 4 for the one-point case.

For the two-point case, however, we cannot keep Assumption 4′, as $f$ is no longer differentiable. Instead, we will also use function unbiasedness. In that case, we will not use any additional assumptions on noise variance, as gradient of the smoothed function is already bounded by $G$ as it is Lipschitz and differentiable.

**Assumption 7.** *For all $x \in \mathbb{R}^d$ it holds that $\mathbb{E}_\pi F(x, Z) = f(x)$.*

**Theorem 3.** *Let Assumptions 3 to 6 hold, and consider problem* (4) *solved by Algorithm 1. Then, for any target accuracy $\varepsilon$ and batch size multiplier $B$ (see Tables 2 and 3 for notation), and for a suitable choice of $\gamma, t, p$, the number of oracle calls required to ensure $\mathbb{E}\|x^N - x^*\|^2 \leq \varepsilon$ is bounded by*

$$B \cdot \tilde{\mathcal{O}}\left(\sqrt{\frac{\sqrt{d}G^2}{\mu^2\varepsilon}}\log\frac{1}{\varepsilon} + \frac{d(d + \tau)\sigma_1^2 G^2}{B\mu^4\varepsilon^3} + \frac{dG^2}{B\mu^2\varepsilon}\right) \quad \textit{one-point oracle calls}.$$

We present the following theorems.

**Theorem 3'.** *Assume Assumption 5, 6', 3 and 7 hold, and consider problem* (4) *solved by Algorithm 1. Then, for any target accuracy $\varepsilon$ and batch size multiplier $B$ (see Tables 2 and 3 for notation), and for a suitable choice of $\gamma, t, p$, the number of oracle calls required to ensure $\mathbb{E}\|x^N - x^*\|^2 \leq \varepsilon$ is bounded by*

$$B \cdot \tilde{\mathcal{O}}\left( \sqrt{\frac{\sqrt{d}G^2}{\mu^2\varepsilon}} \log\frac{1}{\varepsilon} + \frac{(d+\tau)G^2}{B\mu^2\varepsilon} \right) \quad \textit{two-point oracle calls}\,.$$

As we can see, there is no dependence on the mixing time as long as it is less then the dimension of the problem. Our results coincide with previous work under i.i.d. noise when applied with $\tau = 1$, as previously claimed in Table 1.

### B.3. Oracle complexity bound

In the main part we focused on *expected* number of oracle calls to achieve accuracy $\varepsilon$. It is a common measure of oracle complexity for many algorithms, but one may ask for a stronger *high-probability* bound. Usually, high-probability bounds easily follow from CLT as number of iterations grow. However, we should be more careful as our batch size distribution depends on the number of iterations $N$. We recall that the batch size $b_i$ comes from truncated log-geometric distribution

$$b_i = \begin{cases} 2^{J_i}l, & 2^{J_i} < M \\ l, & \text{else} \end{cases}, \quad J_i \sim \text{Geom}(1/2)$$

and $M$ depends on the number of iterations $N$ as $M \lesssim \frac{N}{\log N}$. With that, we apply Bernstein's inequality to the sum $S_N = \sum b_i$:

$$P(S_N > \alpha\mathbb{E}[S_N]) \leq \exp\left( -\frac{\alpha^2 N^2[\mathbb{E}b_1]^2}{2N\mathbb{E}[b_1^2] + \frac{2\alpha}{3}MN[\mathbb{E}b_1]} \right) \leq$$

$$\exp\left( -c\frac{\alpha^2 N^2 l^2 (\log M)^2}{NMl^2 + \alpha MNl(\log M)} \right) \leq e^{-c(\log M)^2\alpha}.$$

It shows the subexponential behavior of the normalized deviation from the mean, thus confirming that the expectation is typical in the high-probability sense.

## C. Notations and definitions.

In this section we define the shorthanded notation used in the proof of upper bounds. For general notations and definitions, see Tables 2 and 3.

Markovian error:

$$h(x, Z) := F(x, Z) - f(x) \tag{10}$$

Single sample gradient estimators:

$$\hat{g}_i := d\frac{\hat{F}(x + te_i, Z_i^{(+)}) - \hat{F}(x - te_i, Z_i^{(-)})}{2t}e_i \tag{11}$$

$$\tilde{g}_i := d\frac{F(x + te_i, Z_i^{(+)}) - F(x - te_i, Z_i^{(-)})}{2t}e_i \tag{12}$$

$$\stackrel{(10)}{=} d\frac{f(x + te_i) + h(x + te_i, Z_i^{(+)}) - f(x - te_i) - h(x - te_i, Z_i^{(-)})}{2t}e_i$$

$$g_i := d\frac{f(x + te_i) - f(x - te_i)}{2t}e_i \tag{13}$$

Batched gradient estimators:

$$\hat{g}^j := \hat{g}_{rd}[2^j l] = \frac{1}{2^j l}\sum_{i=1}^{2^j l}\hat{g}_i \tag{14}$$

(Not to be confused with $\hat{g}^k$, which is $\hat{g}_{ml}$ calculated on $k$-th iteration)

$$\tilde{g}^j := \frac{1}{2^j l} \sum_{i=1}^{2^j l} \tilde{g}_i \tag{15}$$

$$g^j := \frac{1}{2^j l} \sum_{i=1}^{2^j l} g_i \tag{16}$$

Directional gradients:

$$\nabla_{e_i} f(x_0) := d\langle \nabla f(x_0), e_i \rangle e_i \tag{17}$$
$$\nabla_{e_i} F_i := d\langle \nabla F(x, Z_i), e_i \rangle e_i \tag{18}$$

Misc:

$$\mathbb{E}_e := \mathbb{E}_{e_1, e_2, \ldots, e_{2^j l}} \tag{19}$$

$\mathbb{E}_Z := \mathbb{E}_{Z_1, Z_2, \ldots, Z_{2^j l}}$, where $Z_1 \sim \xi$ - arbitrary initial distribution on $(\mathsf{Z}, \mathcal{Z})$

$\mathbb{E} := \mathbb{E}_Z \mathbb{E}_e$

$\mathcal{F}_k := \sigma(x^1, x^2, \ldots, x^k)$ - sigma algebra of first $k$ iterations

$\mathbb{E}_k[\cdot] := \mathbb{E}[\cdot | \mathcal{F}_k]$

$$r^N := \frac{1}{\mu}(f(x_f^N) - f(x^*)) + \left\| x^N - x^* \right\|^2 \tag{20}$$

# D. Proofs of one-point results

## D.1. Markov variance reduction

**Lemma 3** (Extended version of Lemma 1). *Let Assumptions 3 and 4(4′) hold. Then for any $n \geq 1$ and $x \in \mathbb{R}^d$ and any initial distribution $\xi$ on $(\mathsf{Z}, \mathcal{Z})$, we have*

$$\mathbb{E}_Z \left[ \left( \frac{1}{n} \sum_{i=1}^{n} \mathbb{E}_{e_i}\left[ h(x + te_i, Z_i)e_i \right] \right)^2 \right] \lesssim \frac{\tau}{dn} \sigma_1^2, \tag{21}$$

$$\mathbb{E}_Z \left[ \left\| \frac{1}{n} \sum_{i=1}^{n} \nabla F(x, Z_i) - \nabla f(x) \right\|^2 \right] \lesssim \frac{\tau}{n} \sigma_2^2, \tag{22}$$

*Proof.* The proof of (22) can be found in Lemma 1 of Beznosikov et al. (2024).

The proof under Assumption 4 relies on the fact that aforementioned Lemma 1 requires just the two following conditions from the stochastic realizations $\nabla F(x, Z_i)$:

$$\begin{cases} \mathbb{E}_\pi \nabla F(x, Z_i) = \nabla f(x) \\ \left\| \nabla F(x, Z_i) - \nabla f(x) \right\|^2 \leq \sigma_2^2 \end{cases}$$

Denote $h_t(x, Z_i) := \mathbb{E}_e\left[ h(x + te, Z_i)e \right]$, $e \sim RS_2^d(1)$.

Thus (21) $\Leftrightarrow \mathbb{E}_Z \left[ \left( \frac{1}{n} \sum_{i=1}^{n} h_t(x, Z_i) \right)^2 \right] \lesssim \frac{\tau}{n}\frac{\sigma_1^2}{d} \Leftrightarrow \begin{cases} \mathbb{E}_\pi h_t(x, Z_i) = 0 \\ \left\| h_t(x, Z_i) \right\|^2 \lesssim \frac{\sigma_1^2}{d} \end{cases}$

Let's prove both of these equations, starting with unbiasedness:

$$\mathbb{E}_\pi h_t(x, Z_i) = \mathbb{E}_\pi \mathbb{E}_e\left[ h(x + te, Z_i)e \right] = \mathbb{E}_e \mathbb{E}_\pi\left[ h(x + te, Z_i)e \right] \stackrel{(4)}{=} \mathbb{E}_e 0 = 0$$

$$
\begin{aligned}
\|h_t(x, Z_i)\|^2 &= \|\mathbb{E}_e\left[h(x + te, Z_i)e\right]\|^2 \\
&= \langle \mathbb{E}_e\left[h(x + te, Z_i)e\right], h_t(x, Z_i)\rangle \\
&\overset{①}{=} \mathbb{E}_e\left[h(x + te, Z_i) \cdot \langle e, h_t(x, Z_i)\rangle\right] \\
&\overset{②}{\leq} \sqrt{\mathbb{E}_e h(x + te, Z_i)^2} \cdot \sqrt{\mathbb{E}_e \langle e, h_t(x, Z_i)\rangle^2} \\
&\overset{(81)}{=} \sqrt{\mathbb{E}_e h(x + te, Z_i)^2} \cdot \sqrt{\frac{1}{d}\|h_t(x, Z_i)\|^2} \\
&\overset{(4)}{\leq} \sqrt{\sigma_1^2} \cdot \sqrt{\frac{1}{d}\|h_t(x, Z_i)\|^2},
\end{aligned}
$$

where ① holds since $h_t(x, Z_i)$ does not depend on $e$, and ② is a Cauchy-Shwartz inequality for the following dot product: $\langle x(e), y(e)\rangle := \mathbb{E}_e\left[x \cdot y\right]$.

To conclude the proof we square the inequality we got:

$$
\|h_t(x, Z_i)\|^2 \leq \frac{\sqrt{\sigma_1^2}}{\sqrt{d}} \cdot \sqrt{\|h_t(x, Z_i)\|^2} \Rightarrow \|h_t(x, Z_i)\|^2 \leq \frac{\sigma_1^2}{d}.
$$

$\square$

### D.2. Properties of smoothed function

The following lemma establishes key properties of the $l_2$-ball smoothed function

**Lemma 4.** *Assume $f$ is convex. Then the following holds for all $x \in \mathbb{R}^d$*

$$
\begin{aligned}
&\textit{If } f \textit{ is L-smooth / G-Lipschitz / } \mu\textit{-strongly convex [Assumptions 1, 2 and 6],} &(23)\\
&\textit{then } f_t \textit{ from (2) is also L-smooth / G-Lipschitz / } \mu\textit{-strongly convex.}
\end{aligned}
$$

$$
\nabla f_t(x) = \mathbb{E}_e\left[g(x)\right], \tag{24}
$$

$$
f_t(x) \geq f(x), \tag{25}
$$

*If $f$ is additionally $G$-Lipschitz:*

$$
f_t(x) \leq f(x) + Gt, \tag{26}
$$

$$
f_t \textit{ is L-smooth with } L = \frac{\sqrt{d}G}{t}, \tag{27}
$$

*If $f$ is additionally $L$-smooth:*

$$
f_t(x) \leq f(x) + Lt^2, \tag{28}
$$

$$
\|\nabla f(x) - \nabla f_t(x)\|^2 \leq L^2 t^2, \tag{29}
$$

$$
\|\nabla f_t(x)\|^2 \geq \tfrac{1}{2}\|\nabla f(x)\|^2 - L^2 t^2. \tag{30}
$$

*Proof.* Proving (23), we start with $G$-Lipschitzness:

$$
\begin{aligned}
|f_t(x) - f_t(y)| &= |\mathbb{E}_r\left[f(x + tr) - f(y + tr)\right]| \\
&\overset{(80)}{\leq} \mathbb{E}_r |f(x + tr) - f(y + tr)| \\
&\overset{(6)}{\leq} \mathbb{E}_r G\|x - y\| = G\|x - y\|.
\end{aligned}
$$

Next, $L$-smoothness is analogous. Finally, $\mu$-strong convexity of $f_t$, (24), (25) and (28) are proven in Lemmas A2-A3 of (Akhavan et al., 2020).

(26) and (27) can be seen in section 4.1 of Gasnikov et al. (2020).

We prove the rest of inequalities in order.

Proof of (29):

$$\begin{aligned}
\|\nabla f(x) - \nabla f_t(x)\|^2 &= \|\nabla f(x) - \mathbb{E}_r \nabla f(x + tr)\|^2 \\
&= \|\mathbb{E}_r \left[\nabla f(x) - \nabla f(x + tr)\right]\|^2 \\
&\stackrel{(80)}{\leq} \mathbb{E}_r \|\nabla f(x) - \nabla f(x + tr)\|^2 \\
&\stackrel{(1)}{\leq} \mathbb{E}_r L^2 t^2 = L^2 t^2.
\end{aligned}$$

Proof of (30):

$$\begin{aligned}
\|\nabla f_t(x)\|^2 &= \|\nabla f(x) + [\nabla f_t(x) - \nabla f(x)]\|^2 \\
&\stackrel{\text{①}}{\geq} \frac{1}{2}\|\nabla f(x)\|^2 - \|\nabla f_t(x) - \nabla f(x)\|^2 \\
&\stackrel{(29)}{\geq} \frac{1}{2}\|\nabla f(x)\|^2 - L^2 t^2,
\end{aligned}$$

where ① uses that $\|a + b\|^2 \geq 1/2\|a\|^2 - \|b\|^2$. $\qquad\qquad\square$

### D.3. Inequalities for gradient approximation

**Lemma 5.** *Assume Assumption 1, Assumption 3 and Assumption 4. Then the following inequalities hold for any initial distribution $\xi$ on $(\mathsf{Z}, \mathcal{Z})$ and for all $x \in \mathbb{R}^d$:*

$$\left\|\hat{g}^j - \tilde{g}^j\right\|^2 \leq \frac{d^2 \Delta^2}{t^2}, \tag{31}$$

$$\mathbb{E}\|\tilde{g}_i - g_i\|^2 \leq \frac{d^2 \sigma_1^2}{t^2}, \tag{32}$$

$$\mathbb{E}\left\|\mathbb{E}_e \left[\tilde{g}^j - g^j\right]\right\|^2 \leq \frac{d C_1 \tau \sigma_1^2}{t^2 2^j l}, \tag{33}$$

$$\mathbb{E}\|g_i - \nabla_{e_i} f\|^2 \leq \frac{d^2 L^2 t^2}{4}, \tag{34}$$

$$\mathbb{E}\left\|\tilde{g}^j - \mathbb{E}_e \tilde{g}^j\right\|^2 \leq \frac{3}{2^j l}\left[\frac{d^2 \sigma_1^2}{t^2} + \frac{d^2 L^2 t^2}{4} + d\|\nabla f\|^2\right], \tag{35}$$

$$\mathbb{E}\left\|\tilde{g}^j - \mathbb{E}_e g^j\right\|^2 \lesssim \frac{d(d+\tau)\sigma_1^2}{t^2 2^j l} + \frac{d^2 L^2 t^2}{2^j l} + \frac{d\|\nabla f\|^2}{2^j l}, \tag{36}$$

$$\mathbb{E}\left\|\hat{g}^j - \nabla f_t\right\|^2 \lesssim \frac{d^2 \Delta^2}{t^2} + \frac{d(d+\tau)\sigma_1^2}{t^2 2^j l} + \frac{d^2 L^2 t^2}{2^j l} + \frac{d\|\nabla f\|^2}{2^j l}, \tag{37}$$

$$\left\|\mathbb{E}\hat{g}^j - \nabla f_t\right\|^2 \leq \frac{2d^2 \Delta^2}{t^2} + \frac{2d C_1 \tau \sigma_1^2}{t^2 2^j l}. \tag{38}$$

$$\tag{39}$$

*Proof.* We prove all estimates one by one, starting with (31):

$$\begin{aligned}
\left\|\hat{g}^j - \tilde{g}^j\right\|^2 &\stackrel{(14),(15)}{=} \left\|\frac{1}{2^j l} \sum_{i=1}^{2^j l} [\hat{g}_i - \tilde{g}_i]\right\|^2 \\
&\stackrel{(11),(12)}{=} \frac{d^2}{4t^2} \left\|\frac{1}{2^j l} \sum_{i=1}^{2^j l} [\hat{F}(x + te_i, Z_i^+) - \hat{F}(x - te_i, Z_i^-) \right. \\
&\qquad\qquad\qquad\qquad \left. - F(x + te_i, Z_i^+) + F(x - te_i, Z_i^-)]e_i\right\|^2 \\
&\stackrel{(9)}{=} \frac{d^2}{4t^2} \left\|\frac{1}{2^j l} \sum_{i=1}^{2^j l} [\Delta(x + te_i) - \Delta(x - te_i)]\, e_i\right\|^2
\end{aligned}$$

$$\overset{(77)}{\leq} \quad \frac{d^2}{4t^2 2^j l} \sum_{i=1}^{2^j l} \left\| \left[ \Delta(x + te_i) - \Delta(x - te_i) \right] e_i \right\|^2$$

$$\overset{\|e_i\|=1}{=} \quad \frac{d^2}{4t^2 2^j l} \sum_{i=1}^{2^j l} \left| \Delta(x + te_i) - \Delta(x - te_i) \right|^2$$

$$\overset{(9)}{\leq} \quad \frac{d^2}{4t^2} 4\Delta^2$$

$$= \quad \frac{d^2 \Delta^2}{t^2}.$$

Proof of (32):

$$\mathbb{E}\|\tilde{g}_i - g_i\|^2 \overset{(12),(13)}{=} \mathbb{E}\left\| d \frac{h(x + te_i, Z_i^+) - h(x - te_i, Z_i^-)}{2t} e_i \right\|^2$$

$$\overset{\|e_i\|=1}{=} \frac{d^2}{4t^2} \mathbb{E}\left[ h(x + te_i, Z_i^+) - h(x - te_i, Z_i^-) \right]^2$$

$$\overset{(77),(4)}{\leq} \frac{d^2 \sigma_1^2}{t^2}.$$

Proof of (33):

$$\mathbb{E}\left\| \mathbb{E}_e \left[ \tilde{g}^j - g^j \right] \right\|^2$$

$$\overset{(12),(16)}{=} \mathbb{E}\left\| \frac{1}{2^j l} \sum_{i=1}^{2^j l} \mathbb{E}_e \left[ d \frac{h(x + te_i, Z_i^+) - h(x - te_i, Z_i^-)}{2t} e_i \right] \right\|^2$$

$$= \frac{d^2}{t^2} \mathbb{E}\left\| \frac{1}{2^j l} \sum_{i=1}^{2^j l} \mathbb{E}_e \left[ \frac{h(x + te_i, Z_i^+)e_i - h(x - te_i, Z_i^-)e_i}{2} \right] \right\|^2$$

$$\overset{(77)}{\leq} \frac{d^2}{t^2} \frac{1}{2} \left[ \mathbb{E}\left\| \frac{1}{2^j l} \sum_{i=1}^{2^j l} \mathbb{E}_e \left[ h(x + te_i, Z_i^+)e_i \right] \right\|^2 + \mathbb{E}\left\| \frac{1}{2^j l} \sum_{i=1}^{2^j l} \mathbb{E}_e \left[ h(x - te_i, Z_i^-)e_i \right] \right\|^2 \right]$$

$$\overset{(21)}{\leq} \frac{d C_1 \tau \sigma_1^2}{t^2 2^j l}.$$

Proof of (34):

$$\mathbb{E}\|g_i - \nabla_{e_i} f\|^2$$

$$\overset{(13),(17)}{=} \mathbb{E}\left\| d \frac{f(x + te_i) - f(x - te_i)}{2t} e_i - d\langle \nabla f(x), e_i \rangle e_i \right\|^2$$

$$= d^2 \mathbb{E}\left| \frac{f(x + te_i) - f(x) + f(x) - f(x - te_i) - 2t\langle \nabla f(x), e_i \rangle}{2t} \right|^2$$

$$= d^2 \mathbb{E}\left| \frac{f(x + te_i) - f(x) - \langle \nabla f(x), te_i \rangle}{2t} + \frac{f(x) - f(x - te_i) + \langle \nabla f(x), -te_i \rangle}{2t} \right|^2$$

$$\overset{①}{\leq} \frac{2d^2}{4t^2} \left( \frac{L^2 t^4}{4} + \frac{L^2 t^4}{4} \right)$$

$$= \frac{d^2 L^2 t^2}{4},$$

where ① uses Assumption 1, (74) and (77).

Proof of (35):

$$\mathbb{E}\left\| \tilde{g}^j - \mathbb{E}_e \tilde{g}^j \right\|^2 \overset{(15)}{=} \mathbb{E}_Z \mathbb{E}_e \left\| \frac{1}{2^j l} \sum_{i=1}^{2^j l} \left[ \tilde{g}_i - \mathbb{E}_{e_i} \tilde{g}_i \right] \right\|^2$$

$$\overset{\text{①}}{=} \quad \mathbb{E}_Z \mathbb{E}_e \frac{1}{2^{2j} l^2} \sum_{i=1}^{2^j l} \|\tilde{g}_i - \mathbb{E}_{e_i} \tilde{g}_i\|^2$$

$$\overset{(78)}{\leq} \quad \frac{1}{2^{2j} l^2} \sum_{i=1}^{2^j l} \mathbb{E}_Z \mathbb{E}_e \|\tilde{g}_i\|^2$$

$$\overset{(77)}{\leq} \quad \frac{3}{2^{2j} l^2} \sum_{i=1}^{2^j l} \mathbb{E} \left[ \|\tilde{g}_i - g_i\|^2 + \|g_i - \nabla_{e_i} f\|^2 + \|\nabla_{e_i} f\|^2 \right]$$

$$\overset{(32),(34),(81)}{\leq} \quad \frac{3}{2^j l} \left[ \frac{d^2 \sigma_1^2}{t^2} + \frac{d^2 L^2 t^2}{4} + d\|\nabla f\|^2 \right],$$

where ① holds, since $\tilde{g}_i$ are independent w.r.t. $e_i$ and $\mathbb{E}_e \left[ \tilde{g}_i - \mathbb{E}_{e_i} [\tilde{g}_i] \right] = 0$.

Proof of (36):

$$\mathbb{E}\|\tilde{g}^j - \mathbb{E}_e g^j\|^2 \quad \overset{(77)}{\leq} \quad 2\mathbb{E} \left[ \|\tilde{g}^j - \mathbb{E}_e \tilde{g}^j\|^2 + \|\mathbb{E}_e \tilde{g}^j - \mathbb{E}_e g^j\|^2 \right]$$

$$\overset{(35),(33)}{\leq} \quad 2 \left[ \frac{3}{2^j l} \left[ \frac{d^2 \sigma_1^2}{t^2} + \frac{d^2 L^2 t^2}{4} + d\|\nabla f\|^2 \right] + \frac{dC_1 \tau \sigma_1^2}{t^2 2^j l} \right]$$

$$\lesssim \quad \frac{d(d+\tau)\sigma_1^2}{t^2 2^j l} + \frac{d^2 L^2 t^2}{2^j l} + \frac{d\|\nabla f\|^2}{2^j l}.$$

Proof of (37):

$$\mathbb{E}\|\hat{g}^j - \nabla f_t\|^2 \quad \overset{(77)}{\leq} \quad 2\mathbb{E} \left[ \|\hat{g}^j - \tilde{g}^j\|^2\| + \|\tilde{g}^j - \mathbb{E}_e g^j\|^2 \right]$$

$$\overset{(31),(36)}{\lesssim} \quad \frac{d^2 \Delta^2}{t^2} + \frac{d(d+\tau)\sigma_1^2}{t^2 2^j l} + \frac{d^2 L^2 t^2}{2^j l} + \frac{d\|\nabla f\|^2}{2^j l}.$$

Proof of (38):

$$\|\mathbb{E}\hat{g}^j - \nabla f_t\|^2 \quad \overset{(77)}{\leq} \quad 2\|\mathbb{E}\hat{g}^j - \mathbb{E}\tilde{g}^j\|^2 + 2\|\mathbb{E}\tilde{g}^j - \nabla f_t\|^2$$

$$\overset{(24)}{=} \quad 2\|\mathbb{E}\hat{g}^j - \mathbb{E}\tilde{g}^j\|^2 + 2\|\mathbb{E}_Z \mathbb{E}_e \tilde{g}^j - \mathbb{E}_e g^j\|^2$$

$$\overset{(80)}{\leq} \quad 2\|\mathbb{E}\hat{g}^j - \mathbb{E}\tilde{g}^j\|^2 + 2\mathbb{E}_Z \|\mathbb{E}_e \tilde{g}^j - \mathbb{E}_e g^j\|^2$$

$$\overset{(31),(33)}{\leq} \quad \frac{2d^2 \Delta^2}{t^2} + \frac{2dC_1 \tau \sigma_1^2}{t^2 2^j l}.$$

$\square$

**Lemma 6.** *Assume Assumption 6, Assumption 2, Assumption 4. Then the following inequalities hold for any initial distribution $\xi$ on $(Z, \mathcal{Z})$ and for all $x \in \mathbb{R}^d$:*

$$\mathbb{E}\|g_i\|^2 \lesssim dG^2, \tag{40}$$

$$\mathbb{E}\|\tilde{g}^j - \mathbb{E}_e \tilde{g}^j\|^2 \lesssim \frac{2}{2^j l} \left[ \frac{d^2 \sigma_1^2}{t^2} + dG^2 \right], \tag{41}$$

$$\mathbb{E}\|\tilde{g}^j - \mathbb{E}_e g^j\|^2 \lesssim \frac{dC_1(d+\tau)\sigma_1^2}{t^2 2^j l} + \frac{dG^2}{2^j l}. \tag{42}$$

$$\mathbb{E}\|\hat{g}^j - \nabla f_t\|^2 \lesssim \frac{d^2 \Delta^2}{t^2} + \frac{dC_1(d+\tau)\sigma_1^2}{t^2 2^j l} + \frac{dG^2}{2^j l}. \tag{43}$$

*Proof.*
Proof of (40):

$$\mathbb{E}\|g_i\|^2 \quad \overset{(11)}{=} \quad \frac{d^2}{4t^2} \mathbb{E}\left|f(x + te_i) - f(x - te_i)\right|^2$$

$$\overset{(77)}{\leq} \quad \frac{d^2}{2t^2}\mathbb{E}\Big[|f(x+te_i) - \mathbb{E}_{e_i}f(x+te_i)|^2 + |\mathbb{E}_{e_i}f(x+te_i) - f(x-te_i)|^2\Big]$$

$$\overset{①}{\leq} \quad \frac{d^2}{t^2}\mathbb{E}\,|f(x+te_i) - \mathbb{E}_{e_i}f(x+te_i)|^2$$

$$\overset{②}{\lesssim} \quad dG^2,$$

where ① uses that the distribution of $e_i$ is symmetric, and
② uses the fact that for $f$ which is $G$-Lipshitz and $e \in RS_2^d(1)$ it holds that $\mathbb{E}[f(e) - \mathbb{E}_e f(e)]^2 \lesssim \frac{G^2}{d}$ [same reasoning as (Shamir, 2017), Lemma 9].

Proof of (41):

$$\mathbb{E}\big\|\tilde{g}^j - \mathbb{E}_e\tilde{g}^j\big\|^2 \quad \overset{①}{\leq} \quad \frac{1}{2^{2j}l^2}\sum_{i=1}^{2^j l}\mathbb{E}_Z\mathbb{E}_e\|\tilde{g}_i\|^2$$

$$\overset{(77)}{\leq} \quad \frac{2}{2^{2j}l^2}\sum_{i=1}^{2^j l}\mathbb{E}\Big[\|\tilde{g}_i - g_i\|^2 + \|g_i\|^2\Big]$$

$$\overset{(32),(40)}{\leq} \quad \frac{2}{2^j l}\left[\frac{d^2\sigma_1^2}{t^2} + dG^2\right],$$

where ① is analogous to (35).
Proof of (42):

$$\mathbb{E}\big\|\tilde{g}^j - \mathbb{E}_e g^j\big\|^2 \quad \overset{(77)}{\leq} \quad 2\mathbb{E}\Big[\big\|\tilde{g}^j - \mathbb{E}_e\tilde{g}^j\big\|^2 + \big\|\mathbb{E}_e\tilde{g}^j - \mathbb{E}_e g^j\big\|^2\Big]$$

$$\overset{(41),(33)}{\leq} \quad 2\left[\frac{2}{2^j l}\left[\frac{d^2\sigma_1^2}{t^2} + dG^2\right] + \frac{dC_1\tau\sigma_1^2}{t^2 2^j l}\right]$$

$$\lesssim \quad \frac{d\,(d+\tau)\,\sigma_1^2}{t^2 2^j l} + \frac{dG^2}{2^j l}.$$

Proof of (43):

$$\mathbb{E}\big\|\hat{g}^j - \nabla f_t\big\|^2 \quad \overset{(77)}{\leq} \quad 2\mathbb{E}\Big[\big\|\hat{g}^j - \tilde{g}^j\big\|^2 + \big\|\mathbb{E}_e\tilde{g}^j - \nabla f_t\big\|^2\Big]$$

$$\overset{(31),(42)}{\lesssim} \quad \frac{d^2\Delta^2}{t^2} + \frac{d\,(d+\tau)\,\sigma_1^2}{t^2 2^j l} + \frac{dG^2}{2^j l}.$$

$$\square$$

**Lemma 7** (Lemma 2). *Let Assumptions 3 and 4 hold. For any initial distribution $\xi$ on $(\mathsf{Z}, \mathcal{Z})$ the gradient estimates $\hat{g}_{ml}$ satisfy $\mathbb{E}[\hat{g}_{ml}] = \mathbb{E}\big[\hat{g}_{rd}\big[2^{\lfloor\log_2 M\rfloor}l\big]\big]$. Moreover,*

$$\|\nabla f_t(x) - \mathbb{E}[\hat{g}_{ml}]\|^2 \lesssim \frac{d^2\Delta^2}{t^2} + \frac{d\tau\sigma_1^2}{t^2 MB}. \tag{44}$$

*Moreover, under assumption Assumption 1*

$$\mathbb{E}[\|\nabla f_t(x) - \hat{g}_{ml}\|^2] \lesssim \frac{d^2\Delta^2}{t^2} + \frac{d\,(d+\tau)\,\sigma_1^2}{t^2 B} + \frac{d^2 L^2 t^2}{B} + \frac{d}{B}\|\nabla f\|^2. \tag{45}$$

*While under assumption Assumption 6*

$$\mathbb{E}[\|\nabla f_t(x) - \hat{g}_{ml}\|^2] \lesssim \frac{d^2\Delta^2}{t^2} + \frac{d\,(d+\tau)\,\sigma_1^2}{t^2 B} + \frac{dG^2}{B}. \tag{46}$$

*Proof.* Recall that $\hat{g}_{ml}$ is a sum of a baseline estimate $\hat{g}_{rd}[l] \overset{(14)}{=} \hat{g}^0$ and a refining term $2^J[\hat{g}^J - \hat{g}^{J-1}]$. To show that $\mathbb{E}[\hat{g}_{ml}] = \mathbb{E}\hat{g}^{\lfloor \log_2 M \rfloor}$, then, we use the law of total expectation:

$$\mathbb{E}[\hat{g}_{ml}] = \mathbb{E}\left[\mathbb{E}_J[\hat{g}_{ml}]\right] = \mathbb{E}[\hat{g}^0] + \sum_{j=1}^{\lfloor \log_2 M \rfloor} \mathbb{P}\{J = j\} \cdot 2^j \mathbb{E}[\hat{g}^j - \hat{g}^{j-1}] \tag{47}$$

$$= \mathbb{E}[\hat{g}^0] + \sum_{j=1}^{\lfloor \log_2 M \rfloor} \mathbb{E}[\hat{g}^j - \hat{g}^{j-1}] = \mathbb{E}\hat{g}^{\lfloor \log_2 M \rfloor}.$$

This immediately helps us prove the statement (44):

$$\|\nabla f_t(x) - \mathbb{E}\hat{g}_{ml}\|^2 = \left\|\nabla f_t(x) - \mathbb{E}\left[\hat{g}^{\lfloor \log_2 M \rfloor}\right]\right\|^2 \overset{(38)}{\leq} \frac{2d^2\Delta^2}{t^2} + \frac{2dC_1\tau\sigma_1^2}{t^2 2^{\lfloor \log_2 M \rfloor}l} \overset{l \geq B}{\sim} \frac{d^2\Delta^2}{t^2} + \frac{d\tau\sigma_1^2}{t^2 MB}.$$

Proving the statement of (45) we also start with total expectation:

$$\mathbb{E}[\|\nabla f(x) - \hat{g}_{ml}\|^2]$$

$$\overset{(77)}{\leq} 2\mathbb{E}[\|\nabla f(x) - \hat{g}^0\|^2] + 2\mathbb{E}[\|\hat{g}_{ml} - \hat{g}^0\|^2]$$

$$= 2\mathbb{E}[\|\nabla f(x) - \hat{g}^0\|^2] + 2\sum_{j=1}^{\lfloor \log_2 M \rfloor} \mathbb{P}\{J = j\} \cdot 4^j \mathbb{E}[\|\hat{g}^j - \hat{g}^{j-1}\|^2]$$

$$= 2\mathbb{E}[\|\nabla f(x) - \hat{g}^0\|^2] + 2\sum_{j=1}^{\lfloor \log_2 M \rfloor} 2^j \mathbb{E}[\|\hat{g}^j - \hat{g}^{j-1}\|^2]$$

$$\overset{①}{=} 2\mathbb{E}[\|\nabla f(x) - \hat{g}^0\|^2] + 2\sum_{j=1}^{\lfloor \log_2 M \rfloor} 2^j \mathbb{E}[\|\tilde{g}^j - \tilde{g}^{j-1}\|^2]$$

$$\overset{(77)}{\leq} 2\mathbb{E}[\|\nabla f(x) - \hat{g}^0\|^2] + 4\sum_{j=1}^{\lfloor \log_2 M \rfloor} 2^j \left(\mathbb{E}\|\tilde{g}^j - \mathbb{E}_e g^j\|^2 + \mathbb{E}\|\mathbb{E}_e g^{j-1} - \tilde{g}^{j-1}\|^2\right)$$

$$\leq 2\mathbb{E}[\|\nabla f(x) - \hat{g}^0\|^2] + 16\sum_{j=0}^{\lfloor \log_2 M \rfloor} 2^j \mathbb{E}[\|\mathbb{E}_e g^j - \tilde{g}^j\|^2]$$

$$\overset{(37),(36)}{\sim} 2\left[\frac{d^2\Delta^2}{t^2} + \frac{d(d+\tau)\sigma_1^2}{t^2 l} + \frac{d^2 L^2 t^2}{l} + \frac{d}{l} \cdot \|\nabla f\|^2\right] +$$

$$16\sum_{j=0}^{\lfloor \log_2 M \rfloor} 2^j \left[\frac{d(d+\tau)\sigma_1^2}{t^2 2^j l} + \frac{d^2 L^2 t^2}{2^j l} + \frac{d\|\nabla f\|^2}{2^j l}\right]$$

$$\overset{l \geq \log_2 M \cdot B}{\sim} 2\left[\frac{d^2\Delta^2}{t^2} + \frac{d(d+\tau)\sigma_1^2}{t^2 B} + \frac{d^2 L^2 t^2}{B} + \frac{d}{B} \cdot \|\nabla f\|^2\right] +$$

$$16\left[\frac{d(d+\tau)\sigma_1^2}{t^2 B} + \frac{d^2 L^2 t^2}{B} + \frac{d\|\nabla f\|^2}{B}\right]$$

$$\lesssim \frac{d^2\Delta^2}{t^2} + \frac{d(d+\tau)\sigma_1^2}{t^2 B} + \frac{d^2 L^2 t^2}{B} + \frac{d}{B}\|\nabla f\|^2,$$

where ① uses that $\hat{g}^j - \hat{g}^{j-1} = \tilde{g}^j - \tilde{g}^{j-1}$, since $\tilde{g}^j - \hat{g}^j \overset{(31)}{=} \tilde{g}^{j-1} - \hat{g}^{j-1}$.

The proof of (46) is exactly the same, replacing (37) and (36) with (43) and (42).

$$\mathbb{E}[\|\nabla f(x) - \hat{g}_{ml}\|^2] \lesssim \frac{d^2\Delta^2}{t^2} + \frac{d(d+\tau)\sigma_1^2}{t^2 B} + \frac{dG^2}{B}.$$

$\square$

### D.4. Proof of Theorem 1

The proof of Theorem 1 requires two technical Lemmas.

**Lemma 8.** *Assume Assumptions 1 and 2. Then for the iterates of Algorithm 1 with $\theta = (p\eta^{-1} - 1)/(\beta p\eta^{-1} - 1)$, $\theta > 0$, $\eta \geq 1$, $p > 0$ and arbitrary $\alpha > 0$ it holds that*

$$
\begin{aligned}
\mathbb{E}_k[\|x^{k+1} - x^*\|^2] \leq & (1 + \alpha p\gamma\eta)(1 - \beta)\|x^k - x^*\|^2 + (1 + \alpha p\gamma\eta)\beta\|x_g^k - x^*\|^2 \\
& + (1 + \alpha p\gamma\eta)(\beta^2 - \beta)\|x^k - x_g^k\|^2 + p^2\eta^2\gamma^2\mathbb{E}_k[\|\hat{g}^k\|^2] \\
& - 2\eta^2\gamma\langle\nabla f(x_g^k), x_g^k + (p\eta^{-1} - 1)x_f^k - \eta^{-1}px^*\rangle \\
& + \frac{p\eta\gamma}{\alpha}\|\mathbb{E}_k[\hat{g}^k] - \nabla f(x_g^k)\|^2.
\end{aligned}
\tag{48}
$$

**Lemma 9.** *Assume Assumptions 1 and 2. Let problem (4) be solved by Algorithm 1. Then for any $u \in \mathbb{R}^d$, we get*

$$
\begin{aligned}
\mathbb{E}_k\left[f(x_f^{k+1})\right] \leq & f(u) - \langle\nabla f(x_g^k), u - x_g^k\rangle - \frac{\mu}{2}\|u - x_g^k\|^2 - \frac{\gamma}{2}\|\nabla f(x_g^k)\|^2 \\
& + \frac{\gamma}{2}\|\mathbb{E}_k[\hat{g}^k] - \nabla f(x_g^k)\|^2 + \frac{L\gamma^2}{2}\mathbb{E}_k[\|\hat{g}^k\|^2].
\end{aligned}
$$

These are proven in Beznosikov et al. (2024) as Lemmas 5 and 6, with a slightly different notation: $\hat{f}$ corresponds to $f$ and $\hat{g}$ to $g$.

**Lemma 10** (stepsize tuning). *Given an optimization error after $N$ iterations bounded by*

$$
r^N \leq \exp(-N\Gamma a)r^0 + \Gamma b
$$

*and an upper bound on stepsize $\Gamma \leq \frac{1}{u}$ there exists a constant stepsize $\Gamma_0 \leq \frac{1}{u}$, such that*

$$
r^N = \tilde{\mathcal{O}}\left(\exp\left(-\frac{Na}{u}\right)r^0 + \frac{b}{aN}\right)
$$

*Equivalently, the number of iterations to get $r^N \lesssim \varepsilon$:*

$$
N = \tilde{\mathcal{O}}\left(\frac{u}{a}\ln\varepsilon^{-1} + \frac{b}{a\varepsilon}\right)
\tag{49}
$$

*Proof.* This setup is a simpler version of the one considered in Section 4 of Stich (2019) and so we will tune $\Gamma$ similarly to their Lemma 2:

$$
\Gamma := \min\left(\frac{\ln\max(2, ar^0N/b)}{aN}, \frac{1}{u}\right)
$$

If $\frac{1}{u} < \frac{\ln\max(2, ar^0N/b)}{aN}$, then $\Gamma := \frac{1}{u}$.

$$
r^N \leq \exp\left(\frac{-Na}{u}\right)r^0 + \frac{b}{u} \leq \exp\left(\frac{-Na}{u}\right)r^0 + \frac{b\ln(\ldots)}{aN} = \tilde{\mathcal{O}}\left(\exp\left(-\frac{Na}{u}\right)r^0 + \frac{b}{aN}\right)
$$

Otherwise $\frac{\ln\max(2, ar^0N/b)}{aN} \leq \frac{1}{u}$ and $\Gamma := \frac{\ln\max(2, ar^0N/b)}{aN}$, with $\Gamma b = \tilde{\mathcal{O}}(\frac{b}{aN})$ immediately.

$$
\exp(-N\Gamma a)r^0 = \exp\left(-\ln\max(2, ar^0N/b)\right) = \frac{1}{\max(2, ar^0N/b)}
$$

If $ar^0N/b > 2$, we also get $\tilde{\mathcal{O}}(\frac{b}{aN})$, else $\frac{1}{2} \leq \frac{b}{aNr^0}$ and we get $\tilde{\mathcal{O}}(\frac{b}{aN})$ as well.

To conclude the proof we should mitigate the fact that the stepsize currently depends on the number of iterations. This can easily be done via a restart procedure which would run the algorithm for $N = 1, 2, 4, \ldots$ iterations with a stepsize $\Gamma(N)$. $\square$

**Theorem 4** (Theorem 1). *Let Assumptions 1 to 4 hold, and consider problem (4) solved by Algorithm 1. Then, for a suitable choice of hidden parameters (with $p \simeq \frac{B}{B+d}$) and arbitrary choice of free parameters (see Table 3), it holds that:*

$$\mathbb{E}r^N \lesssim \exp\left(-\sqrt{\frac{p^2\mu\gamma N^2}{3}}\right)r^0 + \frac{p\sqrt{\gamma}}{\mu^{3/2}}\cdot\left[\sigma_1^2\frac{d(d+\tau)}{t^2B} + t^2\frac{L^2d^2}{B}\right] + \frac{\Delta^2d^2}{\mu^2t^2} + \frac{Lt^2}{\mu}$$

*Moreover, for arbitrary $\varepsilon \gtrsim \frac{d\Delta\sqrt{L}}{\mu^{3/2}}$ and an appropriate choice of $t$ and $\gamma$, the number of oracle calls required to ensure $r^N \lesssim \varepsilon$ is bounded by*

$$B\cdot\tilde{\mathcal{O}}\left(\max\left[1,\frac{d}{B}\right]\sqrt{\frac{L}{\mu}}\log\frac{1}{\varepsilon} + \frac{Ld(d+\tau)\sigma_1^2}{B\mu^3\varepsilon^2}\right)\quad\text{one-point oracle calls}\,.$$

*Proof.* Applying Lemma 9 with $u = x^*$ (for arbitrary $x^*$) and $u = x_f^k$ to $f_t$, we get:

$$\mathbb{E}_k\left[f_t(x_f^{k+1})\right] \leq f_t(x^*) - \langle\nabla f_t(x_g^k), x^* - x_g^k\rangle - \frac{\mu}{2}\|x^* - x_g^k\|^2 - \frac{p\gamma}{2}\|\nabla f_t(x_g^k)\|^2 \tag{50}$$
$$+ \frac{p\gamma}{2}\left\|\mathbb{E}_k[\hat{g}^k] - \nabla f_t(x_g^k)\right\|^2 + \frac{Lp^2\gamma^2}{2}\mathbb{E}_k\left[\|\hat{g}^k\|^2\right],$$

$$\mathbb{E}_k\left[f_t(x_f^{k+1})\right] \leq f_t(x_f^k) - \langle\nabla f_t(x_g^k), x_f^k - x_g^k\rangle - \frac{\mu}{2}\|x_f^k - x_g^k\|^2 - \frac{p\gamma}{2}\|\nabla f_t(x_g^k)\|^2 \tag{51}$$
$$+ \frac{p\gamma}{2}\left\|\mathbb{E}_k[\hat{g}^k] - \nabla f_t(x_g^k)\right\|^2 + \frac{Lp^2\gamma^2}{2}\mathbb{E}_k\left[\|\hat{g}^k\|^2\right].$$

Combining $2p\gamma\eta\cdot(50) + 2\gamma\eta(\eta - p)\cdot(51) + (48)$ we get:

$$\mathbb{E}_k[\|x^{k+1} - x^*\|^2 + 2\gamma\eta^2 f_t(x_f^{k+1})]$$
$$\leq (1 + \alpha p\gamma\eta)(1 - \beta)\|x^k - x^*\|^2 + (1 + \alpha p\gamma\eta)\beta\|x_g^k - x^*\|^2$$
$$+ (1 + \alpha p\gamma\eta)(\beta^2 - \beta)\|x^k - x_g^k\|^2 - 2\eta^2\gamma\langle\nabla f_t(x_g^k), x_g^k + (p\eta^{-1} - 1)x_f^k - \eta^{-1}px^*\rangle$$
$$+ p^2\eta^2\gamma^2\mathbb{E}_k\left[\|\hat{g}^k\|^2\right] + \frac{p\eta\gamma}{\alpha}\|\mathbb{E}_k[\hat{g}^k] - \nabla f_t(x_g^k)\|^2$$
$$+ 2p\gamma\eta\Big(f_t(x^*) - \langle\nabla f_t(x_g^k), x^* - x_g^k\rangle - \frac{\mu}{2}\|x^* - x_g^k\|^2 - \frac{p\gamma}{2}\|\nabla f_t(x_g^k)\|^2$$
$$+ \frac{p\gamma}{2}\|\mathbb{E}_k[\hat{g}^k] - \nabla f_t(x_g^k)\|^2 + \frac{Lp^2\gamma^2}{2}\mathbb{E}_k\left[\|\hat{g}^k\|^2\right]\Big)$$
$$+ 2\gamma\eta(\eta - p)\Big(f_t(x_f^k) - \langle\nabla f_t(x_g^k), x_f^k - x_g^k\rangle - \frac{\mu}{2}\|x_f^k - x_g^k\|^2 - \frac{p\gamma}{2}\|\nabla f_t(x_g^k)\|^2$$
$$+ \frac{p\gamma}{2}\|\mathbb{E}_k[\hat{g}^k] - \nabla f_t(x_g^k)\|^2 + \frac{Lp^2\gamma^2}{2}\mathbb{E}_k\left[\|\hat{g}^k\|^2\right]\Big)$$
$$= (1 + \alpha p\gamma\eta)(1 - \beta)\|x^k - x^*\|^2 + 2\gamma\eta(\eta - p)f_t(x_f^k) + 2p\gamma\eta f_t(x^*)$$
$$+ ((1 + \alpha p\gamma\eta)\beta - p\gamma\eta\mu)\|x_g^k - x^*\|^2$$
$$+ (1 + \alpha p\gamma\eta)(\beta^2 - \beta)\|x^k - x_g^k\|^2 - p\gamma^2\eta^2\|\nabla f_t(x_g^k)\|^2$$
$$+ \left(\frac{p\eta\gamma}{\alpha} + p\gamma^2\eta^2\right)\|\mathbb{E}_k[\hat{g}^k] - \nabla f_t(x_g^k)\|^2 + (p^2\eta^2\gamma^2 + p^2\gamma^3\eta^2 L)\mathbb{E}_k\left[\|\hat{g}^k\|^2\right]$$
$$\overset{(76)}{\leq} (1 + \alpha p\gamma\eta)(1 - \beta)\|x^k - x^*\|^2 + 2\gamma\eta(\eta - p)f_t(x_f^k) + 2p\gamma\eta f_t(x^*)$$
$$+ ((1 + \alpha p\gamma\eta)\beta - p\gamma\eta\mu)\|x_g^k - x^*\|^2$$
$$+ (1 + \alpha p\gamma\eta)(\beta^2 - \beta)\|x^k - x_g^k\|^2 - p\gamma^2\eta^2\|\nabla f_t(x_g^k)\|^2$$
$$+ p\eta\gamma\left(\frac{1}{\alpha} + \gamma\eta\right)\|\mathbb{E}_k[\hat{g}^k] - \nabla f_t(x_g^k)\|^2 + 2p^2\eta^2\gamma^2(1 + \gamma L)\mathbb{E}_k\left[\|\hat{g}^k - \nabla f_t(x_g^k)\|^2\right]$$

$$+ 2p^2\eta^2\gamma^2 (1 + \gamma L) \, \mathbb{E}_k \left[ \underbrace{\left\| \nabla f_t(x_g^k) \right\|^2}_{x_g^k \in \mathcal{F}_k} \right].$$

Choosing $\alpha = \frac{\beta}{2p\eta\gamma}$ gives:

$$\beta = \sqrt{4p^2\mu\gamma/3} \overset{\gamma \le \frac{3}{4L}}{\le} \sqrt{p^2\mu/L} < 1,$$

$$(1 + \alpha p\eta\gamma)(1 - \beta) = \left(1 + \frac{\beta}{2}\right)(1 - \beta) \le \left(1 - \frac{\beta}{2}\right),$$

$$((1 + \alpha p\eta\gamma)\beta - p\mu\gamma\eta) = \left(\beta + \frac{\beta^2}{2} - p\mu\gamma\eta\right) \overset{\beta < 1}{<} \left(\frac{3\beta}{2} - p\mu\gamma\eta\right) \overset{p\mu\gamma\eta = 3\beta/2}{\le} 0.$$

Thus:

$$\mathbb{E}_k \left[ \left\| x^{k+1} - x^* \right\|^2 + 2\gamma\eta^2 f_t(x_f^{k+1}) \right]$$

$$\le (1 - \beta/2)\left\| x^k - x^* \right\|^2 + 2\gamma\eta \, (\eta - p) \, f_t(x_f^k) + 2p\gamma\eta f_t(x^*)$$

$$+ p\eta^2\gamma^2 \, (1 + 2p/\beta) \left\| \mathbb{E}_k[\hat{g}^k] - \nabla f_t(x_g^k) \right\|^2$$

$$+ 2p^2\eta^2\gamma^2 \, (1 + \gamma L) \, \mathbb{E}_k \left[ \left\| \hat{g}^k - \nabla f_t(x_g^k) \right\|^2 \right]$$

$$- p\gamma^2\eta^2(1 - 2p(1 + \gamma L))\left\| \nabla f_t(x_g^k) \right\|^2.$$

Subtracting $2\gamma\eta^2 f_t(x^*)$ from both sides, we get:

$$\mathbb{E}_k \left[ \left\| x^{k+1} - x^* \right\|^2 + 2\gamma\eta^2(f_t(x_f^{k+1}) - f_t(x^*)) \right]$$

$$\le (1 - \beta/2) \left\| x^k - x^* \right\|^2 + (1 - p/\eta) \cdot 2\gamma\eta^2(f_t(x_f^k) - f_t(x^*))$$

$$+ p\eta^2\gamma^2 \, (1 + 2p/\beta) \left\| \mathbb{E}_k[\hat{g}^k] - \nabla f_t(x_g^k) \right\|^2$$

$$+ 2p^2\eta^2\gamma^2 \, (1 + \gamma L) \, \mathbb{E}_k \left[ \left\| \hat{g}^k - \nabla f_t(x_g^k) \right\|^2 \right]$$

$$- p\gamma^2\eta^2(1 - 2p(1 + \gamma L))\left\| \nabla f_t(x_g^k) \right\|^2$$

$$\overset{\beta/2 = p/\eta}{=} (1 - \beta/2) \left[ \left\| x^k - x^* \right\|^2 + 2\gamma\eta^2(f_t(x_f^k) - f_t(x^*)) \right]$$

$$+ p\eta^2\gamma^2 \, (1 + 2p/\beta) \left\| \mathbb{E}_k[\hat{g}^k] - \nabla f_t(x_g^k) \right\|^2$$

$$+ 2p^2\eta^2\gamma^2 \, (1 + \gamma L) \, \mathbb{E}_k \left[ \left\| \hat{g}^k - \nabla f_t(x_g^k) \right\|^2 \right]$$

$$- p\gamma^2\eta^2(1 - 2p(1 + \gamma L))\left\| \nabla f_t(x_g^k) \right\|^2.$$

Applying Lemma 7, one can obtain:

$$\mathbb{E}_k \left[ \left\| x^{k+1} - x^* \right\|^2 + 2\gamma\eta^2(f_t(x_f^{k+1}) - f_t(x^*)) \right]$$

$$\lesssim (1 - \beta/2) \left[ \left\| x^k - x^* \right\|^2 + 2\gamma\eta^2(f_t(x_f^k) - f_t(x^*)) \right]$$

$$+ p\eta^2\gamma^2 \, (1 + 2p/\beta) \cdot \left[ \frac{d^2\Delta^2}{t^2} + \frac{d\tau\sigma_1^2}{t^2 M B} \right]$$

$$+ 2p^2\eta^2\gamma^2 \, (1 + \gamma L) \cdot \left[ \frac{d^2\Delta^2}{t^2} + \frac{d \, (d + \tau) \, \sigma_1^2}{t^2 B} + \frac{d^2 L^2 t^2}{B} + \frac{d}{B}\left\| \nabla f(x_g^k) \right\|^2 \right]$$

$$- p\gamma^2\eta^2(1 - 2p(1 + \gamma L))\left\| \nabla f_t(x_g^k) \right\|^2$$

$$= \left/ \frac{1}{M} = p(1 + 2p/\beta)^{-1} \right/$$

$$= (1 - \beta/2) \left[ \left\| x^k - x^* \right\|^2 + 2\gamma\eta^2 (f_t(x_f^k) - f_t(x^*)) \right]$$

$$+ p^2\eta^2\gamma^2 \cdot \left[ \frac{d^2\Delta^2 M}{t^2} + \frac{d\tau\sigma_1^2}{t^2 B} \right]$$

$$+ 2p^2\eta^2\gamma^2 (1 + \gamma L) \cdot \left[ \frac{d^2\Delta^2}{t^2} + \frac{d(d+\tau)\sigma_1^2}{t^2 B} + \frac{d^2 L^2 t^2}{B} + \frac{d}{B} \left\| \nabla f(x_g^k) \right\|^2 \right]$$

$$- p\gamma^2\eta^2 (1 - 2p(1 + \gamma L)) \left\| \nabla f_t(x_g^k) \right\|^2$$

$$\overset{(30)}{\lesssim} (1 - \beta/2) \left[ \left\| x^k - x^* \right\|^2 + 2\gamma\eta^2 (f_t(x_f^k) - f_t(x^*)) \right]$$

$$+ \Delta^2 \cdot \left[ \frac{p^2\eta^2\gamma^2 d^2 M + p^2\eta^2\gamma^2 (1 + \gamma L)d^2}{t^2} \right]$$

$$+ \left\| \nabla f_t(x_g^k) \right\|^2 \cdot \left[ p^2\eta^2\gamma^2 (1 + \gamma L)\frac{d}{B} - p\gamma^2\eta^2 (1 - 2p(1 + \gamma L)) \right]$$

$$+ \sigma_1^2 \cdot \left[ \frac{p^2\eta^2\gamma^2 d\tau + p^2\eta^2\gamma^2 (1 + \gamma L)d(d+\tau)}{t^2 B} \right]$$

$$+ \frac{t^2}{B} \cdot p^2\eta^2\gamma^2 (1 + \gamma L)L^2(d^2 + d)$$

$$\overset{\gamma L < 1}{\lesssim} (1 - \beta/2) \left[ \left\| x^k - x^* \right\|^2 + 2\gamma\eta^2 (f_t(x_f^k) - f_t(x^*)) \right]$$

$$+ p^2\eta^2\gamma^2 \cdot \left[ \sigma_1^2 \frac{d(d+\tau)}{t^2 B} + t^2 \frac{L^2 d^2}{B} + \Delta^2 \frac{d^2 M}{t^2} \right]$$

$$+ \left\| \nabla f(x_g^k) \right\|^2 \cdot p\gamma^2\eta^2 \underbrace{\left[ -1 + p(1 + \gamma L)\left( 1 + \frac{d}{B} \right) \right]}_{=0 \text{ for } p \simeq \frac{B}{B+d}}$$

$$\overset{p\eta\gamma = 3\beta/(2\mu)}{\lesssim} (1 - \beta/2) \left[ \left\| x^k - x^* \right\|^2 + 2\gamma\eta^2 (f_t(x_f^k) - f_t(x^*)) \right]$$

$$+ \frac{\beta^2}{\mu^2} \cdot \left[ \sigma_1^2 \frac{d(d+\tau)}{t^2 B} + t^2 \frac{L^2 d^2}{B} + \Delta^2 \frac{d^2 M}{t^2} \right].$$

Finally, we perform the recursion and substitute $\beta = \sqrt{4p^2\mu\gamma/3}$, $\eta = \sqrt{\frac{3}{\mu\gamma}}$, $r_t^N = \left\| x^N - x^* \right\|^2 + \frac{1}{\mu}(f_t(x_f^N) - f_t(x^*))$:

$$\mathbb{E} r_t^N \lesssim \left( 1 - \sqrt{\frac{p^2\mu\gamma}{3}} \right)^N r_t^0$$

$$+ \frac{\beta}{\mu^2} \cdot \left[ \sigma_1^2 \frac{d(d+\tau)}{t^2 B} + t^2 \frac{L^2 d^2}{B} + \Delta^2 \frac{d^2 M}{t^2} \right]$$

$$\lesssim \exp\left( -\sqrt{\frac{p^2\mu\gamma N^2}{3}} \right) r_t^0$$

$$+ \frac{p\sqrt{\gamma}}{\mu^{3/2}} \cdot \left[ \sigma_1^2 \frac{d(d+\tau)}{t^2 B} + t^2 \frac{L^2 d^2}{B} + \Delta^2 \frac{d^2 M}{t^2} \right]$$

$$\overset{①}{\lesssim} \exp\left( -\sqrt{\frac{p^2\mu\gamma N^2}{3}} \right) r_t^0$$

$$+ \frac{p\sqrt{\gamma}}{\mu^{3/2}} \cdot \left[ \sigma_1^2 \frac{d(d+\tau)}{t^2 B} + t^2 \frac{L^2 d^2}{B} \right]$$

$$+ \frac{\Delta^2 d^2}{\mu^2 t^2},$$

where ① uses that $M \simeq \frac{1}{p}\left(1 + \frac{1}{\sqrt{\mu\gamma}}\right) \Rightarrow Mp\sqrt{\gamma} \simeq \sqrt{\gamma} + \frac{1}{\sqrt{\mu}} \leq \frac{1}{\sqrt{L}} + \frac{1}{\sqrt{\mu}} \lesssim \frac{1}{\sqrt{\mu}}$

Recall that $x^*$ is arbitrary. Therefore by setting $x^* = \arg\min f(x)$, we may bound the error for non-smoothed $f$:

$$
\begin{aligned}
r^N &= \|x^N - x^*\|^2 + \frac{6}{\mu}(f(x_f^N) - f(x^*)) \\
&= \|x^N - x^*\|^2 + \frac{6}{\mu}(\underbrace{f(x_f^N) - f_t(x_f^N)}_{\leq 0 \ (28)} \underbrace{-f(x^*) + f_t(x^*)}_{\leq Lt^2 \ (28)}) + \frac{6}{\mu}(f_t(x_f^N) - f_t(x^*)) \\
&\leq r_t^N + 6\frac{Lt^2}{\mu}
\end{aligned}
$$

Thus we get

$$
\begin{aligned}
\mathbb{E}r^N &\lesssim \exp\left(-\sqrt{\frac{p^2 \mu \gamma N^2}{3}}\right) r^0 \\
&\quad + \frac{p\sqrt{\gamma}}{\mu^{3/2}} \cdot \left[\sigma_1^2 \frac{d(d+\tau)}{t^2 B} + t^2 \frac{L^2 d^2}{B}\right] \\
&\quad + \frac{\Delta^2 d^2}{\mu^2 t^2} + \frac{Lt^2}{\mu}
\end{aligned}
$$

To finish the analysis we need to define $t$ and $\gamma$, as well as the tolerable level of noise $\Delta$. Currently we are left with an expression of form:

$$\mathbb{E}r^N \lesssim \exp(-N\Gamma a)r^0 + \Gamma b + c, \Gamma \leq \frac{1}{u}$$

with

$$
\begin{aligned}
\Gamma &= \sqrt{\gamma} \\
u &\simeq \sqrt{L} \\
a &\simeq p\sqrt{\mu} \\
b &\simeq \frac{p}{\mu^{3/2}} \cdot \left[\sigma_1^2 \frac{d(d+\tau)}{t^2 B} + t^2 \frac{L^2 d^2}{B}\right] \\
c &= \frac{\Delta^2 d^2}{\mu^2 t^2} + \frac{Lt^2}{\mu}
\end{aligned}
$$

To get $c \lesssim \varepsilon$ we have to bound $t$:

$$\frac{d\Delta}{\mu\sqrt{\varepsilon}} \lesssim t \lesssim \frac{\sqrt{\mu\varepsilon}}{\sqrt{L}}$$

Thus we bound the adversarial noise $\varepsilon \gtrsim \frac{d\Delta\sqrt{L}}{\mu^{3/2}} \Leftrightarrow \Delta \lesssim \frac{\varepsilon\mu^{3/2}}{d\sqrt{L}}$.

Applying Lemma 10, to get $r^N \lesssim \varepsilon$ one would need $N$ iterations:

$$N = \tilde{\mathcal{O}}\left(\frac{1}{p}\sqrt{\frac{L}{\mu}}\log\frac{1}{\varepsilon} + \frac{d}{B\mu^2\varepsilon}\left[\frac{(d+\tau)\sigma_1^2}{t^2} + L^2 t^2 d\right]\right) \tag{52}$$

Recalling $p \simeq \frac{B}{B+d}$, as well as setting $t$ to its upper bound, we get the total number of iterations:

$$N = \tilde{\mathcal{O}}\left(\left[1 + \frac{d}{B}\right]\sqrt{\frac{L}{\mu}}\log\frac{1}{\varepsilon} + \frac{Ld(d+\tau)\sigma_1^2}{\mu^3\varepsilon^2 B}\right)$$

Finally, as noted in Section 3.1, each $\hat{g}_{ml}$ uses $\tilde{\mathcal{O}}(B)$ oracle calls, thus the oracle complexity is:

$$B \cdot \tilde{\mathcal{O}}\left(\max\left[1, \frac{d}{B}\right]\sqrt{\frac{L}{\mu}}\log\frac{1}{\varepsilon} + \frac{Ld\left(d + \tau\right)\sigma_1^2}{B\mu^3\varepsilon^2}\right) \quad \text{one-point oracle calls .}$$

$\square$

## D.5. Proof of Theorem 3

**Theorem 5** (Theorem 3). *Let Assumptions 2 to 4 and 6 hold, and consider problem* (4) *solved by Algorithm 1. Then, for a suitable choice of hidden parameters (with $p \simeq 1$) and arbitrary choice of free parameters (see Table 3), it holds that:*

$$\mathbb{E}r^N \lesssim \exp\left(-\sqrt{\frac{\mu\gamma N^2}{3}}\right)r^0 + \frac{\sqrt{\gamma}}{\mu^{3/2}} \cdot \left[\sigma_1^2\frac{dC_1\left(d + \tau\right)}{t^2 B} + \frac{G^2 d}{B}\right] + \frac{\Delta^2 d^2}{\mu^2 t^2} + \frac{Gt}{\mu}$$

*Moreover, for arbitrary $\varepsilon \gtrsim \left[\frac{d\Delta G}{\mu^2}\right]^{2/3}$ and an appropriate choice of $t$ and $\gamma$, the number of oracle calls required to ensure $r^N \lesssim \varepsilon$ is bounded by*

$$B \cdot \tilde{\mathcal{O}}\left[\sqrt{\frac{\sqrt{d}G^2}{\mu^2\varepsilon}}\log\frac{1}{\varepsilon} + \frac{d\left(d + \tau\right)\sigma_1^2 G^2}{\mu^4\varepsilon^3 B} + \frac{G^2 d}{B\mu^2\varepsilon}\right] \quad \text{one-point oracle calls .}$$

*Proof.* The proof is almost identical to the smooth case. The difference is we use (46) instead of (45). With that $p \simeq 1$ is enough, as the term with $\left\|\nabla f(x_g^k)\right\|$ no longer exists. Additionally, $\frac{d^2 L^2 t^2}{B} \to \frac{G^2 d}{B}$. Finally, we may use Lemma 9 as smoothed function is indeed smooth (27).

$$\mathbb{E}r^N \lesssim \exp\left(-\sqrt{\frac{p^2\mu\gamma N^2}{3}}\right)r_t^0$$

$$+ \frac{p\sqrt{\gamma}}{\mu^{3/2}} \cdot \left[\sigma_1^2\frac{dC_1\left(d + \tau\right)}{t^2 B} + \frac{G^2 d}{B}\right]$$

$$+ \underbrace{\frac{\Delta^2 d^2}{\mu^2 t^2} + \frac{Gt}{\mu}}_{(26)}$$

$$\overset{p \simeq 1}{\simeq} \exp\left(-\sqrt{\frac{\mu\gamma N^2}{3}}\right)r_t^0$$

$$+ \frac{\sqrt{\gamma}}{\mu^{3/2}} \cdot \left[\sigma_1^2\frac{dC_1\left(d + \tau\right)}{t^2 B} + \frac{G^2 d}{B}\right]$$

$$+ \frac{\Delta^2 d^2}{\mu^2 t^2} + \frac{Gt}{\mu}$$

Applying Lemma 10 with:

$$\Gamma = \sqrt{\gamma}$$

$$u \simeq \sqrt{L} \overset{(27)}{\simeq} \sqrt{\frac{\sqrt{d}G}{t}}$$

$$a = \sqrt{\mu}$$

$$b = \frac{1}{\mu^{3/2}} \cdot \left[\sigma_1^2\frac{dC_1\left(d + \tau\right)}{t^2 B} + \frac{G^2 d}{B}\right]$$

We get that $r^N \lesssim \varepsilon$ takes $N$ iterations:

$$N = \tilde{\mathcal{O}}\left(\sqrt{\frac{\sqrt{d}G}{t\mu}}\log\frac{1}{\varepsilon} + \frac{d}{B\mu^2\varepsilon}\left[\frac{(d+\tau)\,\sigma_1^2}{t^2} + G^2\right]\right).$$

To get $c \lesssim \varepsilon$ we have to bound $t$:

$$\frac{d\Delta}{\mu\sqrt{\varepsilon}} \lesssim t \lesssim \frac{\mu\varepsilon}{G}$$

Thus we bound the adversarial noise $\varepsilon \gtrsim \left[\frac{d\Delta G}{\mu^2}\right]^{2/3} \Leftrightarrow \Delta \lesssim \frac{\varepsilon^{3/2}\mu^2}{dG}$.

Substituting $L = \frac{\sqrt{d}G}{t}$, as well as setting $t$ to its upper bound, we get the total number of iterations:

$$N = \tilde{\mathcal{O}}\left(\sqrt{\frac{\sqrt{d}G^2}{\mu^2\varepsilon}}\log\frac{1}{\varepsilon} + \frac{d\,(d+\tau)\,\sigma_1^2 G^2}{\mu^4\varepsilon^3 B} + \frac{G^2 d}{B\mu^2\varepsilon}\right).$$

And the oracle complexity:

$$B \cdot \tilde{\mathcal{O}}\left(\sqrt{\frac{\sqrt{d}G^2}{\mu^2\varepsilon}}\log\frac{1}{\varepsilon} + \frac{d\,(d+\tau)\,\sigma_1^2 G^2}{\mu^4\varepsilon^3 B} + \frac{G^2 d}{B\mu^2\varepsilon}\right) \quad \text{one-point oracle calls}\,.$$

$\square$

# E. Proofs of two-point results

The proofs for one- and two- point feedback will functionally differ only in Lemma 5 and Lemma 7, while the rest of the machinery will be reused.

## E.1. Inequalities for gradient approximation

**Lemma 5′.** *Assume Assumption 1, Assumption 3 and Assumption 4. Then the following inequalities hold for any initial distribution $\xi$ on $(\mathsf{Z}, \mathcal{Z})$ and for all $x \in \mathbb{R}^d$:*

$$\mathbb{E}[\|\tilde{g}_i - \nabla_{e_i} F_i\|^2] \le \frac{L^2 d^2 t^2}{4}\,, \tag{53}$$

$$\mathbb{E}\|\nabla_{e_i} F_i - \nabla_{e_i} f\|^2 \le d\sigma_2^2\,, \tag{54}$$

$$\mathbb{E}\left\|\tilde{g}^j - \mathbb{E}_e \tilde{g}^j\right\|^2 \le \frac{3}{2^{jl}}\left[d^2 t^2 L^2/4 + d\sigma_2^2 + d\|\nabla f\|^2\right]\,, \tag{55}$$

$$\mathbb{E}_Z\left\|\mathbb{E}_e \tilde{g}^j - \nabla f_t\right\|^2 \lesssim \frac{\tau}{2^{jl}}\sigma_2^2\,, \tag{56}$$

$$\left\|\mathbb{E}\hat{g}^j - \nabla f_t\right\|^2 \lesssim \frac{d^2\Delta^2}{t^2} + \frac{\tau}{2^{jl}}\sigma_2^2\,, \tag{57}$$

$$\mathbb{E}\left\|\tilde{g}^j - \nabla f_t\right\|^2 \lesssim \frac{d^2 t^2 L^2}{2^{jl}} + \frac{d+\tau}{2^{jl}}\sigma_2^2 + \frac{d}{2^{jl}}\|\nabla f\|^2\,. \tag{58}$$

$$\mathbb{E}\left\|\hat{g}^j - \nabla f_t\right\|^2 \lesssim \frac{d^2\Delta^2}{t^2} + \frac{d^2 t^2 L^2}{2^{jl}} + \frac{d+\tau}{2^{jl}}\sigma_2^2 + \frac{d}{2^{jl}}\|\nabla f\|^2\,. \tag{59}$$

*Proof.*
We prove all estimates one by one, starting with (53):

$$\mathbb{E}[\|\tilde{g}_i - \nabla_{e_i} F_i\|^2]$$

$$\overset{(12),(18)}{=} d^2\mathbb{E}\left[\left\|\frac{F(x + te_i, Z_i) - F(x - te_i, Z_i)}{2t}e_i - \langle\nabla F(x, Z_i), e_i\rangle e_i\right\|^2\right]$$

$$\overset{(1')+(74)}{\le} \frac{L^2 d^2 t^2}{4}\,.$$

Proof of (54):

$$\mathbb{E}\|\nabla_{e_i} F_i - \nabla_{e_i} f\|^2 = \mathbb{E}_Z \mathbb{E}_{e_i}\|\nabla_{e_i} F_i - \nabla_{e_i} f\|^2 \overset{(17),(81)}{=} d\mathbb{E}_Z\|\nabla F_i - \nabla f\|^2 \overset{(4')}{\leq} d\sigma_2^2.$$

Proof of (55):

$$\mathbb{E}\big\|\tilde{g}^j - \mathbb{E}_e \tilde{g}^j\big\|^2 \overset{\text{like (35)}}{\leq} \frac{1}{2^{2j} l^2} \sum_{i=1}^{2^j l} \mathbb{E}\|\tilde{g}_i\|^2$$

$$\overset{(77)}{\leq} \frac{3}{2^{2j} l^2} \sum_{i=1}^{2^j l} \Bigg[ \mathbb{E}\Big[\|\tilde{g}_i - \nabla_{e_i} F_i\|^2\Big] +$$

$$\mathbb{E}[\|\nabla_{e_i} F_i - \nabla_{e_i} f\|^2] + \mathbb{E}[\|\nabla_{e_i} f\|^2]\Bigg]$$

$$\overset{(53),(54),(81)}{=} \frac{3}{2^j l}\left[d^2 t^2 L^2/4 + d\sigma_2^2 + d\|\nabla f\|^2\right].$$

Proof of (56):

$$\mathbb{E}_Z\big\|\mathbb{E}_e \tilde{g}^j - \nabla f_t\big\|^2 \overset{(15)}{=} \mathbb{E}_Z\left\|\mathbb{E}_e\left[\frac{1}{2^j l}\sum_{i=1}^{2^j l}\tilde{g}_i\right] - \nabla f_t\right\|^2$$

$$\overset{(24)}{\leq} \mathbb{E}_Z\left\|\frac{1}{2^j l}\sum_{i=1}^{2^j l}\nabla F_t(x, Z_i) - \nabla f_t(x)\right\|^2$$

$$\overset{①}{\lesssim} \frac{\tau}{2^j l}\sigma_2^2,$$

where ① uses (22) for $\nabla F_t, \nabla f_t$. Let us verify that Assumption $4'$ holds:

Unbiasedness:

$$\mathbb{E}_\pi \nabla F_t(x, Z) = \mathbb{E}_\pi \nabla[\mathbb{E}_r F(x + tr, Z)] =$$
$$\mathbb{E}_\pi \mathbb{E}_r \nabla F(x + tr, Z) = \mathbb{E}_r \mathbb{E}_\pi \nabla F(x + tr, Z) = \mathbb{E}_r \nabla f(x + tr) = \nabla f_t(x).$$

Variance:

$$\|\nabla F_t(x, Z) - \nabla f_t(x)\|^2 = \|\mathbb{E}_r \nabla F(x + tr, Z) - \nabla f(x + tr)\|^2 \overset{(80)}{\leq}$$
$$\mathbb{E}_r \|\nabla F(x + tr, Z) - \nabla f(x + tr)\|^2 \overset{(4')}{\leq} \mathbb{E}_r \sigma_2^2 = \sigma_2^2.$$

Proof of (57):

$$\big\|\mathbb{E}\hat{g}^j - \nabla f_t\big\|^2 \overset{(77)}{\leq} 2\big\|\mathbb{E}\hat{g}^j - \mathbb{E}\tilde{g}^j\big\|^2 + 2\|\mathbb{E}\tilde{g}^j - \nabla f_t\|^2$$

$$\overset{(80)}{\leq} 2\big\|\mathbb{E}\hat{g}^j - \mathbb{E}\tilde{g}^j\big\|^2 + 2\mathbb{E}_Z\|\mathbb{E}_e\tilde{g}^j - \nabla f_t\|^2$$

$$\overset{(31),(56)}{\lesssim} \frac{d^2\Delta^2}{t^2} + \frac{\tau}{2^j l}\sigma_2^2.$$

Proof of (58):

$$\mathbb{E}\big\|\tilde{g}^j - \nabla f_t\big\|^2 \overset{(77)}{\leq} 2\mathbb{E}\big\|\tilde{g}^j - \mathbb{E}_e\tilde{g}^j\big\|^2 + 2\mathbb{E}\|\mathbb{E}_e\tilde{g}^j - \nabla f_t\|^2$$

$$\overset{(55),(56)}{\lesssim} \frac{1}{2^j l}\left[d^2 t^2 L^2 + d\sigma_2^2 + d\|\nabla f\|^2\right] + \frac{\tau}{2^j l}\sigma_2^2$$

$$\lesssim \quad \frac{d^2 t^2 L^2}{2^j l} + \frac{d+\tau}{2^j l}\sigma_2^2 + \frac{d}{2^j l}\|\nabla f\|^2.$$

Proof of (59):

$$
\begin{aligned}
\mathbb{E}\big\|\hat{g}^j - \nabla f_t\big\|^2 \quad &\overset{(77)}{\leq} \quad 2\mathbb{E}\big\|\hat{g}^j - \tilde{g}^j\big\|^2 + 2\mathbb{E}\big\|\tilde{g}^j - \nabla f_t\big\|^2 \\
&\overset{(31),(58)}{\lesssim} \quad \frac{d^2\Delta^2}{t^2} + \frac{d^2 t^2 L^2}{2^j l} + \frac{d+\tau}{2^j l}\sigma_2^2 + \frac{d}{2^j l}\|\nabla f\|^2.
\end{aligned}
$$

$\square$

**Lemma 11.** *Assume Assumption 6′, Assumption 2, Assumption 7. Then the following inequalities hold for any initial distribution $\xi$ on $(\mathsf{Z}, \mathcal{Z})$ and for all $x \in \mathbb{R}^d$:*

$$\mathbb{E}\|\tilde{g}_i\|^2 \lesssim dG^2, \tag{60}$$

$$\mathbb{E}\big\|\tilde{g}^j - \mathbb{E}_e \tilde{g}^j\big\|^2 \leq \frac{dG^2}{2^j l}, \tag{61}$$

$$\mathbb{E}\big\|\mathbb{E}_e \tilde{g}^j - \nabla f_t\big\|^2 \leq \frac{4C_1 \tau G^2}{2^j l}, \tag{62}$$

$$\big\|\mathbb{E}\hat{g}^j - \nabla f_t\big\|^2 \lesssim \frac{d^2\Delta^2}{t^2} + \frac{\tau G^2}{2^j l}, \tag{63}$$

$$\mathbb{E}\big\|\hat{g}^j - \nabla f_t\big\|^2 \lesssim \frac{d^2\Delta^2}{t^2} + \frac{(d+\tau)G^2}{2^j l}. \tag{64}$$

*Proof.*
Proof of (60):

$$\mathbb{E}\|\tilde{g}_i\|^2 \overset{(11)}{=} \frac{d^2}{4t^2}\mathbb{E}\left|F(x+te_i, Z_i) - F(x-te_i, Z_i)\right|^2 \overset{\text{like (40)}}{\lesssim} dG^2.$$

Proof of (61):

$$\mathbb{E}\big\|\tilde{g}^j - \mathbb{E}_e \tilde{g}^j\big\|^2 \overset{\text{like (35)}}{\leq} \frac{1}{2^{2j}l^2}\sum_{i=1}^{2^j l}\mathbb{E}_Z \mathbb{E}_e \|\tilde{g}_i\|^2 \overset{(60)}{\leq} \frac{dG^2}{2^j l}.$$

Proof of (62):

$$
\begin{aligned}
\mathbb{E}\big\|\mathbb{E}_e \tilde{g}^j - \nabla f_t\big\|^2 \quad &\overset{(15)}{=} \quad \mathbb{E}\left\|\mathbb{E}_e\left[\frac{1}{2^j l}\sum_{i=1}^{2^j l}\tilde{g}_i\right] - \nabla f_t\right\|^2 \\
&\overset{(24)}{=} \quad \mathbb{E}\left\|\mathbb{E}_e\left[\frac{1}{2^j l}\sum_{i=1}^{2^j l}\nabla F_t(x, Z_i)\right] - \nabla f_t\right\|^2 \\
&\overset{\textcircled{1}}{\leq} \quad \frac{4C_1 \tau G^2}{2^j l},
\end{aligned}
$$

where ① uses (22) with $\sigma_2^2 = 4G^2$. Let us verify that Assumption 4′ holds:

Unbiasedness:

$$
\begin{aligned}
\mathbb{E}_Z[\nabla F_t(x, Z)] \quad &\overset{(24)}{=} \quad \mathbb{E}_Z \mathbb{E}_e\left[d\frac{F(x+te, Z) - F(x-te, Z)}{2t}e\right] \\
&= \quad \mathbb{E}_e \mathbb{E}_Z\left[d\frac{F(x+te, Z) - F(x-te, Z)}{2t}e\right] \\
&\overset{(7)}{=} \quad \mathbb{E}_e\left[d\frac{f(x+te) - f(x-te)}{2t}e\right] \overset{(24)}{=} \nabla f_t(x).
\end{aligned}
$$

Variance: $\|\nabla F_t(x, Z) - \nabla f_t(x)\| \overset{(77)}{\leq} 2\|\nabla F_t(x, Z)\|^2 + 2\|\nabla f_t(x)\|^2 \overset{②}{\leq} 4G^2,$

where ② uses that from Lemma 4 the smoothed $f_t$ and $F_t$ are differentiable, $G$-Lipshitz and thus have norm of their gradients bounded by $G$.

Proof of (63):

$$\left\|\mathbb{E}\hat{g}^j - \nabla f_t\right\|^2 \overset{(77),(80)}{\leq} 2\left[\mathbb{E}\left\|\hat{g}^j - \tilde{g}^j\right\|^2 + \mathbb{E}_Z\|\mathbb{E}_e\tilde{g}^j - \nabla f_t\|^2\right]$$

$$\overset{(31),(61)}{\lesssim} \frac{d^2\Delta^2}{t^2} + \frac{\tau G^2}{2^j l}.$$

Proof of (64):

$$\mathbb{E}\left\|\hat{g}^j - \nabla f_t\right\|^2 \overset{(77)}{\leq} 3\mathbb{E}\left[\left\|\hat{g}^j - \tilde{g}^j\right\|^2 + \|\tilde{g}^j - \mathbb{E}_e\tilde{g}^j\|^2 + \|\mathbb{E}_e\tilde{g}^j - \nabla f_t\|^2\right]$$

$$\overset{(31),(61),(62)}{\lesssim} \frac{d^2\Delta^2}{t^2} + \frac{(d+\tau)G^2}{2^j l}.$$

$\square$

**Lemma 7'.** *Let Assumptions 3 and 4' hold. For any initial distribution $\xi$ on $(\mathsf{Z}, \mathcal{Z})$ the following inequalities hold:*

*Under Assumption 1:*

$$\mathbb{E}[\|\nabla f_t(x) - \hat{g}_{ml}\|^2] \lesssim \frac{d^2\Delta^2}{t^2} + \frac{d^2t^2L^2}{B} + \frac{d+\tau}{B}\sigma_2^2 + \frac{d}{B}\|\nabla f\|^2. \tag{65}$$

$$\|\nabla f_t(x) - \mathbb{E}[\hat{g}_{ml}]\|^2 \lesssim \frac{d^2\Delta^2}{t^2} + \frac{\tau}{MB}\sigma_2^2. \tag{66}$$

*Under Assumption 6:*

$$\mathbb{E}[\|\nabla f_t(x) - \hat{g}_{ml}\|^2] \lesssim \frac{d^2\Delta^2}{t^2} + \frac{(d+\tau)G^2}{2^j l}. \tag{67}$$

$$\|\nabla f_t(x) - \mathbb{E}[\hat{g}_{ml}]\|^2 \lesssim \frac{d^2\Delta^2}{t^2} + \frac{\tau}{MB}G^2. \tag{68}$$

*Proof.* The proof is almost identical to Lemma 7, so we will leave the calculations only.

Proof of (66):

$$\|\nabla f_t(x) - \mathbb{E}[\hat{g}_{ml}]\|^2 \overset{(47)}{=} \left\|\nabla f_t(x) - \mathbb{E}\left[\hat{g}^{\lfloor\log_2 M\rfloor}\right]\right\|^2$$

$$\overset{(57)}{\lesssim} \frac{d^2\Delta^2}{t^2} + \frac{\tau}{MB}\sigma_2^2.$$

Proof of (68):

$$\|\nabla f_t(x) - \mathbb{E}[\hat{g}_{ml}]\|^2 \overset{(47)}{=} \left\|\nabla f_t(x) - \mathbb{E}\left[\hat{g}^{\lfloor\log_2 M\rfloor}\right]\right\|^2$$

$$\overset{(63)}{\lesssim} \frac{d^2\Delta^2}{t^2} + \frac{\tau}{MB}G^2.$$

Proof of (65):

$$\mathbb{E}[\|\nabla f_t(x) - \hat{g}_{ml}\|^2]$$

$$\leq 2\mathbb{E}[\left\|\nabla f_t(x) - \hat{g}^0\right\|^2] + 2\sum_{j=1}^{\lfloor\log_2 M\rfloor} 2^j\mathbb{E}[\|\tilde{g}^j - \tilde{g}^{j-1}\|^2]$$

$$\leq 2\mathbb{E}[\left\|\nabla f_t(x) - \hat{g}^0\right\|^2] + 4\sum_{j=1}^{\lfloor\log_2 M\rfloor} 2^j\left(\mathbb{E}\|\tilde{g}^j - \nabla f_t(x)\|^2 + \mathbb{E}\|\nabla f_t(x) - \tilde{g}^{j-1}\|^2\right)$$

$$\leq \quad 2\mathbb{E}[\|\nabla f_t(x) - \hat{g}^0\|^2] + 16 \sum_{j=0}^{\lfloor \log_2 M \rfloor} 2^j \mathbb{E}[\|\nabla f_t(x) - \hat{g}^j\|^2]$$

$$\overset{(59),(58)}{\lesssim} \quad \frac{d^2\Delta^2}{t^2} + \frac{d^2 t^2 L^2}{B} + \frac{d+\tau}{B}\sigma_2^2 + \frac{d}{B}\|\nabla f\|^2.$$

Proof of (67):

$$\mathbb{E}[\|\nabla f_t(x) - \hat{g}_{ml}\|^2] \overset{(64)}{\lesssim} \frac{d^2\Delta^2}{t^2} + \frac{(d+\tau)G^2}{2^j l}.$$

$\square$

### E.2. Proof of Theorem 1$'$

**Theorem 1$'$.** *Let Assumptions 1$'$ to 4$'$ hold, and consider problem (4) solved by Algorithm 1. Then, for a suitable choice of hidden parameters (with $p \simeq \frac{B}{B+d}$) and arbitrary choice of free parameters (see Table 3), it holds that:*

$$\mathbb{E}r^N \lesssim \exp\left(-\sqrt{\frac{p^2\mu\gamma N^2}{3}}\right) r^0 + \frac{p\sqrt{\gamma}}{\mu^{3/2}} \cdot \left[\sigma_2^2 \frac{d+\tau}{B} + t^2 \frac{L^2 d^2}{B}\right] + \frac{\Delta^2 d^2}{\mu^2 t^2} + \frac{Lt^2}{\mu}$$

*Moreover, for arbitrary $\varepsilon \gtrsim \frac{d\Delta\sqrt{L}}{\mu^{3/2}}$ and an appropriate choice of $t$ and $\gamma$, the number of oracle calls required to ensure $r^N \lesssim \varepsilon$ is bounded by*

$$B \cdot \tilde{\mathcal{O}}\left(\max\left[1, \frac{d}{B}\right]\sqrt{\frac{L}{\mu}}\log\frac{1}{\varepsilon} + \frac{(d+\tau)\sigma_2^2}{B\mu^2\varepsilon}\right) \quad \textit{two-point oracle calls}.$$

*Proof.* Replacing Lemma 7 with Lemma 7$'$ in the proof of Theorem 1, we get:

$$\mathbb{E}r^N \lesssim \exp\left(-\sqrt{\frac{p^2\mu\gamma N^2}{3}}\right) r^0$$

$$+ \frac{p\sqrt{\gamma}}{\mu^{3/2}} \cdot \left[\sigma_2^2 \frac{d+\tau}{B} + t^2 \frac{L^2 d^2}{B}\right]$$

$$+ \frac{\Delta^2 d^2}{\mu^2 t^2} + \frac{Lt^2}{\mu}$$

Applying Lemma 10 with:

$$\Gamma = \sqrt{\gamma}$$
$$u \simeq \sqrt{L}$$
$$a \simeq p\sqrt{\mu}$$
$$b \simeq \frac{p}{\mu^{3/2}} \cdot \left[\sigma_2^2 \frac{d+\tau}{B} + t^2 \frac{L^2 d^2}{B}\right]$$

We get that $r^N \lesssim \varepsilon$ takes $N$ iterations:

$$N = \tilde{\mathcal{O}}\left(\frac{1}{p}\sqrt{\frac{L}{\mu}}\log\frac{1}{\varepsilon} + \frac{1}{B\mu^2\varepsilon}\left[(d+\tau)\sigma_2^2 + L^2 t^2 d^2\right]\right)$$

Bounds on $\Delta$, $t$ and $p$ are inherited: $\varepsilon \gtrsim \frac{d\Delta\sqrt{L}}{\mu^{3/2}} \Leftrightarrow \Delta \lesssim \frac{\varepsilon\mu^{3/2}}{d\sqrt{L}}$, $t \simeq \frac{\sqrt{\mu\varepsilon}}{\sqrt{L}}$, $p \simeq \frac{B}{B+d}$. Thus the total number of iterations is:

$$\tilde{\mathcal{O}}\left(\left[1 + \frac{d}{B}\right]\sqrt{\frac{L}{\mu}}\log\frac{1}{\varepsilon} + \frac{(d+\tau)\sigma_2^2}{B\mu^2\varepsilon}\right)$$

Finally, the oracle complexity is:

$$B \cdot \tilde{\mathcal{O}} \left( \max\left[1, \frac{d}{B}\right] \sqrt{\frac{L}{\mu}} \log \frac{1}{\varepsilon} + \frac{(d + \tau)\sigma_2^2}{B\mu^2\varepsilon} \right) \quad \text{two-point oracle calls}.$$

$\square$

### E.3. Proof of Theorem 3′

**Theorem 3′.** *Let Assumptions 1′ to 4′ hold, and consider problem* (4) *solved by Algorithm 1. Then, for a suitable choice of hidden parameters (with $p \simeq 1$) and arbitrary choice of free parameters (see Table 3), it holds that:*

$$\mathbb{E}r^N \lesssim \exp\left(-\sqrt{\frac{p^2\mu\gamma N^2}{3}}\right) r^0 + \frac{p\sqrt{\gamma}}{\mu^{3/2}} \cdot G^2 \frac{d + \tau}{B} + \frac{\Delta^2 d^2}{\mu^2 t^2} + \frac{Gt}{\mu}$$

*Moreover, for arbitrary $\varepsilon \gtrsim \frac{d\Delta\sqrt{L}}{\mu^{3/2}}$ and an appropriate choice of $t$ and $\gamma$, the number of oracle calls required to ensure $r^N \lesssim \varepsilon$ is bounded by*

$$B \cdot \tilde{\mathcal{O}} \left( \sqrt{\frac{\sqrt{d}G^2}{\mu^2\varepsilon}} \log \frac{1}{\varepsilon} + \frac{(d + \tau)G^2}{B\mu^2\varepsilon} \right) \quad \text{two-point oracle calls}.$$

*Proof.* Replacing (65) and (66) with (67) and (68) in the proof of the smooth case we get:

$$\mathbb{E}r^N \lesssim \exp\left(-\sqrt{\frac{p^2\mu\gamma N^2}{3}}\right) r^0$$

$$+ \frac{p\sqrt{\gamma}}{\mu^{3/2}} \cdot G^2 \frac{d + \tau}{B}$$

$$+ \frac{\Delta^2 d^2}{\mu^2 t^2} + \frac{Gt}{\mu}$$

Applying Lemma 10 with:

$$\Gamma = \sqrt{\gamma}$$

$$u \simeq \sqrt{L} \overset{Lemma\ 4}{\simeq} \sqrt{\frac{\sqrt{d}G}{t}}$$

$$a \simeq p\sqrt{\mu}$$

$$b \simeq \frac{p(d + \tau)G^2}{\mu^{3/2}B}$$

We get that $r^N \lesssim \varepsilon$ takes $N$ iterations:

$$N = \tilde{\mathcal{O}} \left( \sqrt{\frac{\sqrt{d}G}{t\mu}} \log \frac{1}{\varepsilon} + \frac{(d + \tau)G^2}{B\mu^2\varepsilon} \right)$$

Bounds on $\Delta$, $t$ and $p$ are inherited: $\varepsilon \gtrsim \left[\frac{d\Delta G}{\mu^2}\right]^{2/3} \Leftrightarrow \Delta \lesssim \frac{\varepsilon^{3/2}\mu^2}{dG}$, $t \simeq \frac{\mu\varepsilon}{G}$, $p \simeq 1$. Thus the total number of iterations is:

$$\tilde{\mathcal{O}} \left( \sqrt{\frac{\sqrt{d}G^2}{\mu^2\varepsilon}} \log \frac{1}{\varepsilon} + \frac{(d + \tau)G^2}{B\mu^2\varepsilon} \right)$$

Finally, the oracle complexity is:

$$B \cdot \tilde{\mathcal{O}} \left( \sqrt{\frac{\sqrt{d}G^2}{\mu^2 \varepsilon}} \log \frac{1}{\varepsilon} + \frac{(d+\tau)G^2}{B\mu^2 \varepsilon} \right) \quad \text{two-point oracle calls .}$$

$\square$

# F. Lower Bounds

## F.1. Main theorems

First, we introduce the results that confirm the optimality of our analysis with a second moment bounds. By this we mean that we check

$$\mathbb{E}_\pi |F(x, Z) - f(x)|^2 < \sigma_1^2$$

instead of Assumption 4 and

$$\mathbb{E}_\pi \|\nabla F(x, Z) - \nabla f(x)\|^2 < \sigma_2^2$$

instead of Assumption 4'.

Then, we show how to use clipping technique in the construction of the hard instance problems to preserve the lower bounds up to logarithmic factors.

Our main results here are the following two theorems. They show theoretical optimality of our method and analysis in both one-point and two-point regimes.

**Theorem 6** (one-point feedback). *For any (possibly randomized) algorithm that solves the problem* (4)*, there exists a function $f$ that satisfies Assumptions 1 to 4, s.t.*

$$\mathbb{E}\|\hat{x}_N - x^*\|^2 \gtrsim \frac{\sqrt{d(d+\tau)}\sigma_1^2}{\mu\sqrt{N}} \quad \text{as } N \to \infty.$$

*Consequently, to get to the $\varepsilon$-neighborhood of the solution with one-point feedback the algorithm needs at least*

$$N = \Omega \left( \frac{d(d+\tau)\sigma_1^2}{\mu^2 \varepsilon^2} \right) \quad \text{one-point oracle calls.}$$

**Theorem 7** (two-point feedback). *For any (possibly randomized) algorithm that solves the problem* (4)*, there exists a function $f$ that satisfies Assumptions 1' to 4', s.t.*

$$\mathbb{E}\|\hat{x}_N - x^*\|^2 \gtrsim \frac{(d+\tau)\sigma_2^2}{\mu^2 N} \quad \text{as } N \to \infty.$$

*Consequently, to get to the $\varepsilon$-neighborhood of the solution with two-point feedback one needs at least*

$$N = \Omega \left( \frac{(d+\tau)\sigma_2^2}{\mu^2 \varepsilon} \right) \quad \text{two-point oracle calls.}$$

We note that due to the two-part structure of the optimal rates, it is natural to prove both parts separately in a regime where the part becomes dominant. We introduce those regimes:

- $\tau \geq d$ — high-correlation regime

- $\tau \leq d$ — high-dimensional regime

Next, we summarize the lower bounds that we claim to hold in each regime:

*Table 4.* Strongly convex case, lower bounds

| | **high-correlation** | | **high-dimensional** | |
|---|---|---|---|---|
| **ZO 1-point** | $\dfrac{d\tau\sigma_1^2}{\mu^2\varepsilon^2}$ | (**New**, Theorem 8) | $\dfrac{d^2\sigma_1^2}{\mu^2\varepsilon^2}$ | Akhavan et al. (2024) (our Theorem 9) |
| **ZO 2-point** | $\dfrac{\tau\sigma_2^2}{\mu^2\varepsilon}$ | Beznosikov et al. (2024) (even for **FO**) | $\dfrac{d\sigma_2^2}{\mu^2\varepsilon}$ | Duchi et al. (2015) (our Theorem 10) |

It becomes obvious that only 1 out of 4 bounds depend on dimension and mixing time simultaneously. For other cases, we can use existing constructions which deal with mixing and zero-order information separately and adapt them to our assumptions. Combining all four bounds, we come up with tight lower bounds in both one-point and two-point settings. Let us discuss the important related results.

Akhavan et al. (2024) work with a special case of one-point feedback when noise variables do not depend on query points — this makes their lower bound applicable to our case. The only factor they do not consider is $\sigma_1^2$, which, however, appears from their proof if used with scaled Gaussian noise, as well as additional $\mu^2$ factor; see our Theorem 9 for the result. In the work of Beznosikov et al. (2024), a first-order oracle is considered, but the hard instance problem is a 1-dimensional quadratic problem, which makes first-order and zero-order information equivalent. Duchi et al. (2015) consider a general convex case of a two-point setting and provide a tight lower bound. However, their proof can be translated for strongly convex problems using the trick of adding a common quadratic part to each of the linear functions from the hard-to-distinguish family. For a more formal reduction, see Theorem 10.

Finally, we provide a, to the best of our knowledge, novel lower bound in one-point feedback and high-correlation regime.

**Theorem 8** (one-point, high-correlation). *Under the conditions of Theorem 6 the following bound holds:*

$$\mathbb{E}\|\hat{x}_N - x^*\|^2 \gtrsim \frac{\sqrt{d\tau\sigma_1^2}}{\mu\sqrt{N}}.$$

*Proof.* Let's consider family of functions

$$f_\omega(x) = \frac{\mu}{2}\|x\|^2 + \langle S(x), \omega \rangle$$

with $\omega \in \{\pm 1\}^d$ and $S : \mathbb{R}^d \to \mathbb{R}^d$ to be chosen later. For the same values of $\omega$, consider zeroth order oracles

$$F_\omega(x, Z) = \frac{\mu}{2}\|x\|^2 + \langle S(x), Z + \omega \rangle = f_\omega(x) + \langle S(x), Z \rangle$$

and discrete-time Markov process with transition probabilities determined by the formula

$$Z_{t+1} = \begin{cases} \xi_{t+1}, & \text{w.p. } 1/\tau, \\ Z_t, & \text{w.p. } 1 - 1/\tau, \end{cases}$$

where $\{\xi_t\}_{t=1}^\infty$ are independent and

$$\xi_t \sim \pi = \mathcal{N}(0, s^2 I_d).$$

With such pick of $Z_t$ it is clear that Assumption 3 is satisfied and

$$\mathbb{E}_\pi F_\omega(x, Z) = f_\omega(x).$$

Now, we will prove that all algorithms fail at distinguishing between $f_\omega$ in a short amount of time. First, note that

$$\|\hat{x} - x_\omega^*\|^2 \geq \frac{1}{4}\|x_{\omega'}^* - x_\omega^*\|^2 \tag{69}$$

where $\omega' = \arg\min_{\tilde{\omega}} \|\hat{x} - x^*_{\tilde{\omega}}\|^2$. We will later bound $\|x^*_{\omega'} - x^*_{\omega}\|$ using Hamming distance $\rho(\omega', \omega)$. But first, we bound the distance itself.

Applying Assouad's Lemma (Tsybakov, 2009) we get

$$\max_{\omega} \mathbb{E}_{\omega} \rho(\omega', \omega) \geq \frac{d}{2} \left( 1 - \max_{\rho(\omega_1, \omega_2) = 1} \|P_{\omega_1} - P_{\omega_2}\|_{TV} \right) \tag{70}$$

where $P_{\omega}$ denotes joint distribution of outputs of $F_{\omega}$ on sequential queries produced by the algorithm. And $\hat{x}$ is the output of the algorithm after $N$ steps. Now we bound the total variation between neighbouring distributions. First, we use Pinsker's inequality:

$$2\|P_{\omega_1} - P_{\omega_2}\|^2_{TV} \leq D_{KL} \left( \text{Law}(\{\omega_1 + Z_i\}_{i=1}^N), \ \text{Law}(\{\omega_2 + Z_i\}_{i=1}^N) \right) = \Big/$$

Then, using law of total probability, we consider a conditional $KL$-divergence for a fixed set of indices that introduce new samples. The one step $KL$ equals $0$ if it is known that the chain's state did not change. On other steps it equals to the $KL$ between Gaussians with mean $\omega_1$ and $\omega_2$. We group the terms by the number of state switches $k$.

$$\Big/ = \sum_{k=0}^{N} D_{KL} \left( \text{Law}(\{\mathcal{N}(\omega_1, s^2 I)\}_{i=1}^k), \text{Law}(\{\mathcal{N}(\omega_2, s^2 I)\}_{i=1}^k) \right) \mathbb{P}(|\{1 \leq t \leq N : \ Z_t = \xi_t\}| = k)$$

Using $\rho(\omega_1, \omega_2) = 1$, we simplify

$$= \sum_{k=0}^{N} D_{KL} \left( \mathcal{N}(1, s^2 I_k), \ \mathcal{N}(-1, s^2 I_k) \right) \mathbb{P}(|\{1 \leq t \leq N : \ Z_t = \xi_t\}| = k) =$$

$$\sum_{k=0}^{N} \frac{2k}{s^2} \mathbb{P}(|\{1 \leq t \leq N : \ Z_t = \xi_t\}| = k) = \frac{2}{s^2} \sum_{k=0}^{N} k \mathbb{P}(|\{1 \leq t \leq N : \ Z_t = \xi_t\}| = k) =$$

$$\frac{2}{s^2} \mathbb{E}(|\{1 \leq t \leq N : \ Z_t = \xi_t\}|) = \frac{2}{s^2} \sum_{t=1}^{N} \mathbb{E}(I_{Z_t = \xi_t}) = \frac{2N}{s^2 \tau}.$$

Choosing $s^2 = \frac{8N}{\tau}$ we get

$$\|P_{\omega_1} - P_{\omega_2}\|_{TV} \leq \sqrt{\frac{2N\tau}{8N\tau}} = \frac{1}{2}. \tag{71}$$

Now we claim that there exists such pick of $S(x)$, that satisfies Assumptions 1 to 4 and

$$\|x^*_{\omega'} - x^*_{\omega}\|^2 \geq \frac{1}{2} \frac{\sqrt{\frac{\sigma_1^2 \tau}{9}}}{\sqrt{4\mu^2 dN}} \rho(\omega', \omega) = \frac{1}{12} \frac{\sqrt{\sigma_1^2 \tau}}{\sqrt{\mu^2 dN}} \rho(\omega', \omega). \tag{72}$$

Combining (69), (72), (70), (71), we conclude

$$\max_{\omega} \mathbb{E}_{\omega} \|\hat{x} - x^*_{\omega}\|^2 \geq \frac{1}{96} \frac{d\sqrt{\sigma_1^2 \tau}}{\sqrt{\mu^2 dN}} = \frac{1}{96} \frac{\sqrt{d\tau \sigma_1^2}}{\mu \sqrt{N}}.$$

Now we should introduce $S(x)$ and check (72) and Assumptions 1 to 4.

Denote $\delta = \sqrt[4]{\frac{\sigma_1^2 \tau}{\mu^2 dN}}$. Let S(x) be separable and

$$S(x)_i = \frac{\mu}{4} s(x_i) = \frac{\mu}{4} \cdot \begin{cases} 2\delta x_i, & 0 \leq x_i \leq \delta, \\ 3\delta^2 - (x_i - 2\delta)^2, & \delta \leq x_i \leq 2\delta, \\ 3\delta^2, & 2\delta \leq x_i. \end{cases}$$

And $s(x_i)$ is symmetric around zero. It is straightforward to verify that $s(x_i)$ is 2-smooth. To check strong convexity and smoothness of $f_w$ we note that

$$\nabla f_\omega(x) = \mu x + \nabla \langle S(x), \omega \rangle = \mu x + \nabla S(x) \odot \omega,$$

where $\odot$ is a coordinate-wise product. The Lipschitz constant of the second term is bounded

$$\|\nabla S(x) \odot \omega - \nabla S(y) \odot \omega\| = \|\nabla S(x) - \nabla S(y)\| \leq \frac{\mu}{2} \|x - y\|.$$

It means that the strong convexity constant $\mu$ and gradient Lipschitz constant $L$ of the function $f_\omega$ are in range $[\frac{\mu}{2}; \frac{3\mu}{2}]$. Therefore, for a completely rigorous bound, we use $2\mu$ in (72) instead of $\mu$.

It is also straightforward to verify by stationarity condition that $x_\omega^* = -\frac{1}{2}\omega\delta$ and (72) follows. Here we also note that $\|x_\omega^*\|^2 = \frac{1}{2}d\delta^2 < 1$ for big enough $N$, therefore the minimizer of the function lies in the standard unit ball when the desired accuracy is small enough.

Lastly, we need to check the bounded noise assumption (4). With our current setup we can guarantee bounded variance with respect to stationary distribution

$$\mathbb{E}_\pi h^2(x, Z) = \mathbb{E}_\pi \langle S(x), Z \rangle^2 = s^2 \|S(x)\|^2 \leq \frac{9s^2 \mu^2 d\delta^4}{16} = \frac{9N}{2\tau} \frac{\sigma_1^2 \tau}{N} \leq 9\sigma_1^2. \tag{73}$$

Therefore, for a completely rigorous bound, we use $\sigma_1^2/9$ in (72) instead of $\sigma_1^2$. And a proper uniform bound is achieved via clipping, see Section F.2. $\qquad\square$

### F.2. Remarks on clipping

There is, however, another problem we have to deal with — for now there is only a second-moment bound on the noise, just as in other lower bounds used that work with i.i.d. noise instead of Markovian. Tackling uniform boundness of an i.i.d. noise is straightforward — since the noise distribution is Gaussian, we can use tail bounds to clip the noise within $[-\sigma \log N; \sigma \log N]$ for all querying points with probability $1 - o(1/N)$. It gives the desired bounds up to logarithmic factors for Theorems 9 and 10.

However, in the settings of Theorem 8, this trick will not work as the algorithm can deliberately call the oracle at a point that would produce high noise on the next step. To deal with this, we clip the oracle rather then noise.

For some $t > 1$ ($t$ is going to be logarithmic in $N$) we introduce

$$\hat{F}(x, Z) = \max \left( \min_\omega f_\omega(x) - t\sigma_1, \ \min(F(x, Z), \ \max_\omega f_\omega(x) + t\sigma_1) \right).$$

By construction

$$|\hat{F}(x, Z) - \mathbb{E}_\pi \hat{F}(x, Z)|^2 \leq 2t^2\sigma_1^2 + 2|\max_\omega f_\omega(x) - \min_\omega f_\omega(x)|^2 =$$

$$2t^2\sigma_1^2 + 2\|S(x)\|_1^2 \leq 2t\sigma_1^2 + 8d^2\mu^2\delta^4 = 2t^2\sigma_1^2 + \frac{8d^2\sigma_1^2\tau}{N}.$$

Note that for big enough $N$, the second term becomes negligible. Now, the clipping introduces bias of the form

$$|\mathbb{E}_\pi F(x, Z) - \mathbb{E}_\pi \hat{F}(x, Z)| \leq |\mathbb{E}_\pi h(x, Z) I_{h(x,Z) > t\sigma}| \leq$$

$$\leq \int_{t\sigma}^\infty x e^{-\frac{x^2}{2\sigma_1^2}} dx = \sigma_1^2 \int_t^\infty x e^{-\frac{x^2}{2}} dx = \sigma_1^2 e^{-\frac{t^2}{2}}.$$

Choosing $t \sim \log N$ makes this bias superpolinomially small in $N$ i.e. $\lesssim poly(\frac{1}{N})$, making it within an admissible level of adversarial bias $\Delta \lesssim \frac{\varepsilon \mu^{3/2}}{dL}$. This last step, which introduces a bias, can be avoided through a careful adjustments of the Gaussian distributions used in the proof so that the mutual truncation would not result in a change of expected value. This is possible since the total probability mass that is affected by the truncation is exponentially small, therefore the total variation distance remains large after any transformations with this mass.

### F.3. One-point high dimensional regime

An i.i.d. one-point setup is covered by Akhavan et al. (2020), where authors considered a more general case of high-order smoothness of the objective and provided a lower bound for *any* distribution of the additive noise. Our point of view is different – we work with usual smooth functions, consider a limiting behavior when $N \to \infty$ and are free to choose the noise structure. However, we also claim stronger result - our bound shows additional $\mu^2 \sigma_1^2$ scaling and is asymptotically tight, according to the Theorem 1.

**Theorem 9** (one-point, high-dimensional). *Under the conditions of Theorem 6 the following bound holds:*

$$\mathbb{E}\|\hat{x}_N - x^*\|^2 \gtrsim \frac{\sqrt{d^2 \sigma_1^2}}{\mu \sqrt{N}}.$$

*Proof.* Under closer consideration, the proof repeats, simplifies and extends the construction of Akhavan et al. (2020), using our assumptions. But it will be easier for presentation to build on our own notation from Theorem 8.

We consider the same family of functions $f_\omega$, but the noise is i.i.d. and point-independent Gaussian with variance $\sigma_1^2$. This requires redefining $\delta$ and revising (71) and (72). With this noise, we use bound on the $KL$ divergence between neighboring distributions similar to Akhavan et al. (2020, Theorem 6.1). We also use that $I_0 = \frac{1}{2\sigma_1^2}$ for Gaussian distributions. We get

$$D_{KL}(P_{\omega_1}, P_{\omega_2}) \leq \frac{N}{2\sigma_1^2}\|f_{\omega_1} - f_{\omega_2}\|_\infty^2 < \frac{N\mu^2 \delta^4}{2\sigma_1^2}.$$

Redefining $\delta = \sqrt[4]{\frac{\sigma_1^2}{\mu^2 N}}$ we check that (71) holds. The (72) then transforms into

$$\|x_{\omega'}^* - x_\omega^*\|^2 \geq \frac{1}{2}\sqrt{\frac{\sigma_1^2}{4\mu^2 N}}\rho(\omega', \omega).$$

Combining (69), (72), (70), (71), we conclude

$$\max_\omega \mathbb{E}_\omega \|\hat{x} - x_\omega^*\|^2 \geq \frac{1}{16}\frac{d\sqrt{\sigma_1^2}}{\sqrt{\mu^2 N}}.$$

$\square$

### F.4. Two-point high dimensional regime

Theorem 10 below shows a reduction from the lower bound by Duchi et al. (2015) to a strongly convex objectives. Coupled with the clipping technique discussed above, it concludes all the proofs of the section.

**Theorem 10** (two-point, high-dimensional). *Under the conditions of Theorem 7 the following bound holds:*

$$\mathbb{E}\|\hat{x}_N - x^*\|^2 \gtrsim \frac{d\sigma_2^2}{\mu^2 N}.$$

*Proof.* Let's consider family of functions for $v \in \{\pm 1\}^d$

$$f_v(x) = \frac{\mu}{2}\|x\|^2 + \delta\langle x, v\rangle$$

and corresponding oracles

$$F_v(x, Z) = \frac{\mu}{2}\|x\|^2 + \langle x, \delta v + Z \rangle.$$

The noise sequence $Z_i$ is not given any Markovianity, instead we choose it to be i.i.d. $\sim \mathcal{N}(0, s^2 I_d)$. This family readily satisfies Assumptions 1' to 4' with the parameter $\sigma_2^2 \geq \mathbb{E}\|Z\|^2 = ds^2$. Again, here we consider only a second moment bound, as discussed above.

This construction is similar to the one used in a proof by Duchi et al. (2015, Proposition 1), but here we add a deterministic quadratic part, as we work with a strongly convex problems. Therefore, there is always a global minimizer of the function

$$x_v^* = \arg\min f_v(x) = -\frac{\delta}{\mu}v.$$

As usual, we can bound distance to the optima with the Hamming distance between the signs of the estimate and the optima

$$\max_v \mathbb{E}\|\hat{x}_N - x_v^*\|^2 \geq \frac{\delta^2}{\mu^2}\sum_{i=1}^d \mathbb{P}(\text{sign}(\hat{x}_N^i) \neq -\text{sign}(v^i)).$$

Duchi et al. (2015) prove a lower bound on the sum of such probabilities

$$\sum_{i=1}^d \mathbb{P}(\text{sign}(\hat{x}_N^i) \neq -\text{sign}(v^i)) \geq d\left(1 - \sqrt{\frac{2N\delta^2}{ds^2}}\right).$$

This inequality also applies to our set of functions as they differ only by a common deterministic function. Therefore, we get

$$\max_v \mathbb{E}\|\hat{x}_N - x_v^*\|^2 \geq \frac{d\delta^2}{\mu^2}\left(1 - \sqrt{\frac{2N\delta^2}{ds^2}}\right).$$

Choosing $s^2 = \frac{\sigma_2^2}{d}$ and $\delta^2 = \frac{\sigma_2^2}{4N}$ gives the desired result

$$\mathbb{E}\|\hat{x}_N - x^*\|^2 \gtrsim \frac{d\sigma_2^2}{\mu^2 N}.$$

$\square$

## G. Basic Facts

**Lemma 12.** *If $f$ is $L$-smooth in $\mathbb{R}^d$, then for any $x, y \in \mathbb{R}^d$*

$$f(x) - f(y) - \langle \nabla f(y), x - y \rangle \leq \frac{L}{2}\|x - y\|^2. \tag{74}$$

**Lemma 13** (Cauchy Schwartz inequality)**.** *For any $a, b, x_1, \dots, x_n \in \mathbb{R}^d$ and $c > 0$ the following inequalities hold:*

$$2\langle a, b \rangle \leq \frac{\|a\|^2}{c} + c\|b\|^2, \tag{75}$$

$$\|a + b\|^2 \le \left(1 + \frac{1}{c}\right)\|a\|^2 + (1 + c)\|b\|^2, \tag{76}$$

$$\left\|\sum_{i=1}^{n} x_i\right\|^2 \le n \cdot \sum_{i=1}^{n} \|x_i\|^2. \tag{77}$$

**Lemma 14.** *For a random variable $\xi$ with a finite second moment:*

$$\mathbb{E}\|\xi - \mathbb{E}\xi\|^2 \le \mathbb{E}\|\xi\|^2. \tag{78}$$

**Lemma 15** (Jensen's inequality). *If $f$ is a convex function, then for any $n \in \mathbb{N}^*$ and $x_1, \ldots, x_n \in \mathbb{R}^d$ the following inequality holds:*

$$f\left(\frac{1}{n}\sum_{i=1}^{n} x_i\right) \le \frac{1}{n}\sum_{i=1}^{n} f(x_i). \tag{79}$$

*Probabilistic form:*

$$f(\mathbb{E}[X]) \le \mathbb{E}[f(X)].$$

*Applied to $f(X) = \|X\|^2$:*

$$\|\mathbb{E}[X]\|^2 \le \mathbb{E}\left[\|X\|^2\right]. \tag{80}$$

**Lemma 16** (Norm of random projection). *For $e \sim RS_2^d(1)$ the following equality holds:*

$$\mathbb{E}_e\langle v, e\rangle^2 = \|v\|^2 \cdot 1/d. \tag{81}$$

*Proof.*

$$\mathbb{E}\langle v, e\rangle^2 = \|v\|^2 \mathbb{E}\langle v/\|v\|, e\rangle^2 = \|v\|^2 \mathbb{E}\langle(1, 0, \ldots, 0), \tilde{e}\rangle^2 = \|v\|^2 \mathbb{E}[\tilde{e}_1]^2 \overset{①}{=} \|v\|^2 \cdot 1/d,$$

where ① uses $\sum_i \mathbb{E}[\tilde{e}_i]^2 = 1$ and $E[\tilde{e}_1]^2 = \mathbb{E}[\tilde{e}_2]^2 = \ldots$ $\qquad\square$

