# OpenReview forum: "Gradient-Free Approaches is a Key to an Efficient Interaction with Markovian Stochasticity"
_ICML.cc/2026/Conference — ICML 2026 regular_

### Official Review · Reviewer_hBbm · 2026-03-09

**Soundness:** 4
**Presentation:** 3
**Significance:** 4
**Originality:** 3
**Overall Recommendation:** 5
**Confidence:** 3

**Summary:**

This paper proposes and analyzes a novel derivative-free method for stochastic optimization problems with Markovian noise, applicable to both strongly convex smooth and non-smooth cases, using single-point or two-point feedback. Through a stochastic batch scheme, it is demonstrated that when the noise mixing time is less than the problem dimension, the convergence rate is independent of the mixing time, indicating that zeroth-order oracles are superior to expensive first-order oracles in such problems. Finally, matching lower bounds are provided to verify the optimality of the results.

**Compliance With Llm Reviewing Policy:**

Affirmed.

**Final Justification:**

I recommend accepting this article. In this paper, a derivative free method for stochastic optimization problems with Markov noise is proposed and analyzed for two cases of strongly convex smooth and non-smooth. Through the response to the relevant content, although some of its assumptions are somewhat harsh, I think the work is meaningful for a random batch processing scheme.

**Key Questions For Authors:**

（1）Through the example of the lazy Markov chain, it was explained why the final complexity exhibits an additive effect of (d + τ) rather than a multiplicative effect of d·τ. However, it seems that there is a lack of a rigorous lemma or theorem statement in the main text that formalizes this intuitive idea. May I ask if there is a core lemma that strictly proves how d and τ are decoupled in the variance term under random direction gradient estimation? If so, could you clearly indicate its location? If not, could such a core mathematical statement be added to the final version to more clearly support the foundation of the entire paper?
（2）You mentioned in the introduction that "The theory is also numerically validated in Section 3." Unfortunately, this crucial part is not included in the provided materials. Could you please elaborate on the experimental setup? Most importantly, did the experiment clearly demonstrate the expected changes in the algorithm's performance (such as convergence speed) when the mixing time τ changed from being less than d to being greater than d? Did the experimental results replicate the theoretical advantages summarized in Table 1?

**Limitations:**

The author did not adequately address the limitations of the work and the potential negative social impacts. The paper focuses on theoretical analysis but lacks a clear elaboration on the limitations of its own methods in the conclusion or separate sections and does not mention possible negative social effects. It is recommended to add a section titled "Limitations and Future Work" at the end of the paper, covering the following contents: explicitly discussing the applicability of theoretical assumptions in practical problems and the possible performance degradation when these assumptions are not met. Analyzing the sensitivity of the algorithm to hyperparameters and providing heuristic parameter selection suggestions or explanations of the challenges in parameter tuning.

**Strengths And Weaknesses:**

Soundness: The theoretical derivation of the paper is rigorous. It is based on clear and standard assumptions such as strong convexity and smoothness. The proof process is progressive, starting from the basic properties of the gradient estimator, to the analysis of random batching, then to the application of multi-layer Monte Carlo techniques, and finally integrated into the framework of accelerated gradient descent, forming a complete logical chain. At the same time, the application of MLMC and random direction decomposition is reasonable, and finally, it is rigorously quantified in the lemma. Overall, the theoretical part of the paper is very reasonable and rigorous, with an appropriate method selection, demonstrating the author's profound theoretical foundation.

Presentation: The presentation of the paper is excellent, and the core ideas are conveyed clearly and effectively. However, there are a few minor formatting issues that could be addressed. For instance, the diagram in Section 1.1 lacks a figure label, and there is an excessive blank space preceding it. Additionally, the formatting of the references could be improved. Overall, these issues do not hinder the understanding of the content, and the manuscript meets the high standards expected of academic papers.

Significance: Markov noise is widely present in practical applications such as reinforcement learning, online learning, and time series analysis. Meanwhile, zero-order optimization is crucial in scenarios where gradients are unavailable or computational costs are high. Combining these two important and practical fields, the research issues involved inherently have great value.

Originality: The originality of this paper is very high. The traditional view holds that the difficulty in handling Markov noise (τ) and the difficulty in dealing with high-dimensional problems (d) will add up to form a product effect of d·τ. This paper, through meticulous analysis, proves that under the specific framework of zero-order optimization, this effect is additive (d + τ), provided that τ is less than d. This is a completely new and non-obvious insight. Additionally, in terms of technology, it ingeniously combines the MLMC technique for handling Markov noise with the random direction gradient estimation in zero-order optimization. It is not a simple application but deeply analyzes how this combination leads to the scaling of variance terms and bias terms in different ways, thereby achieving decoupling of τ and d.

---

> ### Author Rebuttal · Authors · 2026-03-31
>
> We appreciate Reviewer hBbm’s time, effort, and supportive feedback. Below, we address their questions.
>
> Questions:
> > Q1. ($d$ + $\tau$ formal proof)
>
> While the focus of discussions in sections “Can the mini-batching scheme be improved?” and “Random directions” is on providing intuition, the precise properties of the final MLMC estimator are described in Lemma 2 (line 310), the proof of which can be found in the appendix as Lemma 7 (line 1032).
>
> > Q2. (Experiment clarification)
>
> Section 3 can be found on page 8. The experiment was indeed designed with the goal of displaying the additive dependency of optimization error $\varepsilon$ on the problem dimension $d$ and mixing time $\tau$. Specifically, we optimized a simple quadratic function on a lazy Markov chain while varying $d$, $\tau$ and strength of noise $\sigma_2^2$. The resulting plots $(\tau, d) \to  \varepsilon$ show (noisy) planes, confirming linear, not multiplicative scaling. For details, please see the experiments section (line 426).
>
> > L1. (No discussion of Limitations and Future Work)
>
> You are right, the paper needs to elaborate on Limitations and Future work. We plan to address these concerns in the conclusion, here is the draft:
>
> We study derivative-free stochastic optimization with Markovian noise sequence in one- and two- point settings. We propose a gradient estimation scheme with reduced variance that allows us to achieve additive, not multiplicative complexity w.r.t. problem dimension and mixing time. We establish the optimality of the claimed convergence rate by proving corresponding lower bounds, and validate the theory numerically. Future work might apply our technique to problems with different (non-convex, PL) or more relaxed (ergodicity, bounded variance) assumptions. Finally, a more practically oriented algorithm, which does not require knowledge of mixing time in advance is of interest.
>
> As for hyperparameter tuning, we believe it should be addressed as part of future work, for details please see Q6 by reviewer yMFL.

---

> > ### Author Rebuttal · Reviewer_hBbm · 2026-04-01
> >
> > The author answered my questions in detail. I think the overall quality of this article is quite good.

---

### Official Review · Reviewer_yMFL · 2026-03-11

**Soundness:** 4
**Presentation:** 4
**Significance:** 4
**Originality:** 3
**Overall Recommendation:** 5
**Confidence:** 4

**Summary:**

This paper studies stochastic optimization under Markovian noise with zero-order oracle
access, a combination that has not been previously analyzed. The central and surprising
finding is that the oracle complexity scales as $(d + \tau)$ rather than the naively
expected $d\tau$, where $d$ is the problem dimension and $\tau$ is the mixing time of
the Markov chain. The key technical contribution is a combination of two ideas: (i)
assigning independent random directions to each sample in the mini-batch, which decouples
the variance into a $\tau$-independent term and a direction-averaging term, and (ii)
Multilevel Monte Carlo (MLMC) batching to control bias from temporal correlations. The
paper covers smooth and non-smooth strongly convex objectives under both one-point and
two-point feedback, and provides matching lower bounds confirming the optimality of the
rates up to logarithmic factors.

**Compliance With Llm Reviewing Policy:**

Affirmed.

**Key Questions For Authors:**

1. **On the extra $L$ factor in the one-point smooth rate:** Theorem 1 gives oracle
complexity $\widetilde{O}\!\left(\frac{Ld(d+\tau)\sigma_1^2}{B\mu^3\varepsilon^2}
\right)$, which has an extra $L$ factor compared to the i.i.d. one-point rate
$\widetilde{O}\!\left(\frac{d^2\sigma_1^2}{\mu^3\varepsilon^2}\right)$ from Akhavan
et al. (2024), even when $\tau = 1$. Is this $L$ factor unavoidable in the one-point
smooth setting, or does it reflect a suboptimality in the analysis — for example, in the
choice of $t$? If it is unavoidable, a lower bound capturing it would be needed to
confirm tightness in this regime.

2. **On the necessity of uniform boundedness:** Assumptions 4 and 4' require uniform
bounds $|F(x,Z) - f(x)|^2 \leq \sigma_1^2$ and $\|\nabla F(x,Z) - \nabla f(x)\|^2
\leq \sigma_2^2$ for all $Z \in \mathcal{Z}$, rather than bounded variance in
expectation. Is this necessary for the MLMC bias analysis, or could the results be
extended to second-moment bounds $\mathbb{E}_\pi|F(x,Z) - f(x)|^2 \leq \sigma_1^2$ as
in Even (2023)? Specifically, does Lemma 1 and the MLMC construction in Lemma 7 require
the uniform bound, or only the second moment?

3. **On lower bound coverage in the high-dimensional one-point regime:** Theorem 9 uses
the construction of Akhavan et al. (2020, 2024), which assumes point-independent noise —
strictly weaker than Assumption 4. Can the authors confirm whether the lower bound
$\Omega\!\left(\frac{d^2\sigma_1^2}{\mu^2\varepsilon^2}\right)$ holds under Assumption 4
(query-dependent noise), or only under point-independence? If the latter, the upper and
lower bounds do not share the same noise model in this regime, and the claimed tightness
is not fully established.

4. **On missing related work:** The related work section on the first-order methods under Markovian noise omits the following relevant works:

[AL23] Alacaoglu and Lyu, "Convergence of first-order methods for nonconvex constrained
  optimization with dependent data," ICML 2023.

[PL24] Powell and Lyu, "Stochastic optimization with arbitrary recurrent data sampling,"
  ICML 2024.

These works study first-order optimization under dependent data and recurrent sampling
schemes that generalize or relate to the Markovian setting considered here. Although the focus in the above works is smooth nonconvex constrained minimization and the present work is for strongly convex minimization, the data sampling (or noise) schemes appears to be more general and the assumptions are weaker. Could the authors add some relevant discussions?

5. **On experiments:** The only experiment uses a quadratic objective
$f(x) = \frac{1}{2}\|x\|^2$ with a lazy Gaussian Markov chain — essentially the hard
instance construction from the lower bound proof. This by design matches the theory but
does not demonstrate robustness across problem structures. Could the authors provide
experiments on at least one non-quadratic objective or a Markov chain with a different
mixing structure, e.g., a finite-state irreducible aperiodic chain?

6. **On practical hyperparameter selection:** Table 3 lists many hyperparameters with
complex interdependencies, including $l = (\lfloor \log_2 M \rfloor + 1) \cdot B$ and
$M = \frac{1}{p} + \frac{2}{\beta}$. In the experiments, what specific values were used
for $l$, $M$, $B$, and $\beta$, and how sensitive are the results to these choices in
practice?

**Strengths And Weaknesses:**

## Soundness

The paper is technically careful and the proofs are detailed and well-organized. The
decomposition of variance in Section 2.1 via random directions is clean and the key
intuition — that randomizing directions eliminates one factor of $d$ from the $d\tau$
dependence while MLMC handles bias — is made precise in Lemmas 5 and 7. The lower bound
for the high-correlation regime (Theorem 8) is novel and uses Assouad's lemma with a
careful Markov chain construction; the clipping argument in Section F.2 to convert
second-moment bounds to uniform bounds is standard but correctly handled.

**Some weaknesses**

Despite the strong theoretical contribution, several technical concerns deserve attention:

**1. Extra $L$ factor in the one-point smooth rate.** Theorem 1 gives oracle complexity
scaling as $\frac{Ld(d+\tau)\sigma_1^2}{\mu^3\varepsilon^2}$, which carries an extra
$L$ factor relative to the i.i.d. one-point result
$\frac{d^2\sigma_1^2}{\mu^3\varepsilon^2}$ from Akhavan et al. (2024). This is not a
consequence of Markovian noise — it appears even when $\tau = 1$. The gap arises from
the bias-variance tradeoff in choosing the finite-difference step $t$ and is acknowledged
implicitly but not discussed explicitly. It is unclear whether this $L$ factor is
unavoidable in the one-point smooth setting or reflects a suboptimality in the analysis.

**2. Uniform boundedness is a strong assumption.** Assumptions 4 and 4' require that
$|F(x,Z) - f(x)|^2 \leq \sigma_1^2$ and $\|\nabla F(x,Z) - \nabla f(x)\|^2 \leq
\sigma_2^2$ uniformly for all $Z \in \mathcal{Z}$. This is considerably stronger than the bounded variance assumptions used in most of the ZOO literature with i.i.d. noise, and stronger than what Even (2023) requires even in the first-order Markovian setting. Also in Assumption 2.3 in [AL23] (reference given below), only the boundedness of the one-step conditional expectation of the gradient is assumed. The paper acknowledges this but only briefly, noting that relaxation is possible under stronger structural conditions. A more substantive discussion of whether the uniform bound is necessary for the MLMC analysis would strengthen the paper.

**3. Lower bound coverage is uneven.** Of the four regimes in Table 4, only one bound
(Theorem 8, one-point high-correlation) is truly novel. The other three reuse existing
constructions from Akhavan et al. (2020, 2024) and Duchi et al. (2015) with minor
adaptations. In particular, the one-point high-dimensional lower bound (Theorem 9) relies
on Akhavan et al.'s construction which assumes point-independent noise — a strictly
easier setting than Assumption 4. The claim that the upper and lower bounds match should
therefore be qualified: the match in the high-dimensional one-point regime holds only
under weaker noise assumptions than those of Theorem 1.

**4. Algorithm has many hidden hyperparameters.** Table 3 lists eight hyperparameters
including $J$ (random), $l$, $M$, $B$, $\beta$, $\eta$, $p$, $\theta$, most of which
are set via implicit dependencies described in the table. The practical selection of
these parameters is non-transparent and not discussed in the experiments.


## Presentation

The paper is clearly written with an effective high-level narrative. The hypothesis
diagram in Section 1.1 is an excellent device for motivating the main result. The
intuitive discussion of sparsified sampling and the lazy Markov chain in Section 2.1
makes the key technical insight accessible before the formal development. Notation is
heavy but Tables 2 and 3 are helpful references.

One presentational weakness is that the extra $L$ factor in the one-point smooth rate
is never flagged as a potential gap, leaving the reader to notice it by comparing Table 1
entries. The relationship between the practical batch size multiplier $B$ and the
complexity bounds also warrants a brief practical discussion — as $B \to \infty$ the
stochastic term vanishes but the deterministic term dominates, which is not the typical
operating regime.


## Significance

The paper opens a new and natural research direction by combining two well-studied but
previously separate lines of work — ZOO methods and Markovian optimization. The main
finding, that complexity scales as $(d + \tau)$ rather than $d\tau$, is genuinely
surprising and practically meaningful: it implies that when $\tau \leq d$, switching
from first-order to zero-order oracles incurs no additional cost from Markovian
structure. The connection to reinforcement learning (where first-order information is
unavailable and noise is inherently Markovian) is a strong motivating application.


## Originality

The combination of random directions with MLMC to achieve $(d + \tau)$ scaling is the
core novel technical contribution. The lower bound for the one-point high-correlation
regime (Theorem 8) is new. The remaining lower bounds are adaptations of existing
results. Overall the paper makes a solid original contribution to a previously unstudied
problem setting.

---

> ### Author Rebuttal · Authors · 2026-03-31
>
> We thank the reviewer yMFL for their thoughtful and constructive evaluation of our work. We address the concerns raised below.
>
> > Q1. (On the extra $L$ factor in the one-point smooth rate)
>
> Thank you, that’s a good catch - we misrepresented the upper bound formula in the one-point iid setting. It should include an additional $L$ in the numerator and refer to [1] instead. As a result, our Markovian complexity would match iid in all settings. We note that achieving tight (no $\frac{L}{\mu}$ gap) rates in ZO 1P setting remains an open direction even without Markovian noise, see [1, Theorems 5.1,6.1].
>
> > Q2. (On the necessity of uniform boundedness)
>
> You are right, as long as Lemma 1 is true the proof should work the same. Investigating the sharp theoretical limits for the proposed proof scheme to work is an interesting direction and “Ergodic mirror descent” [2] suggests both Markovianity and uniform boundness of the oracle can be relaxed. Nevertheless, we argue that presenting the first result of this type in our limiting but clear setting is beneficial, as the main insights become easier to understand. We will add this remark in the final version, thank you.
>
> > Q3. (On lower bound coverage in the high-dimensional one-point regime)
>
> Point-independent noise is indeed an “easier” setting for *upper bounds*, compared to noise which is potentially query-dependent, as it permits a narrower class of problems. For the lower bounds, however, that means that any construction in Akhavan et al. (2024) would also fit our broader assumptions.
>
> > Q4. (On missing related work)
>
> Thank you for this remark, we did miss these works during our literature review. We will add them to the discussion of Markovian first-order methods.
>
> > Q5. (On experiments)
>
> The main contribution of the paper is the new gradient estimation technique with reduced variance that allowed us to achieve $(d + \tau)$ oracle complexity. The focus of the experiment is not on the applicability or robustness of the algorithm, but rather on illustrating and numerically confirming the theory. While we believe our theoretical framework to enable practical extensions, we feel they lie outside of scope and should be investigated in future work.
>
> > Q6. (On practical hyperparameter selection)
>
> While all of the parameters may be set with just the knowledge of properties of the problem, in practice we found it easier to manually set $\gamma$ to 1e-3, and $p$ to $d / 30$ (Theorem 1 omits tracking constants for ease of presentation, thus $p$ had to be set manually), while taking the rest of the parameters from Table 3. It is worth reiterating that we did not aim to show immediate practicality of our algorithm with the experiment, but rather to illustrate and numerically confirm the theory. Therefore we did not analyze robustness/adaptivity and focused on the simplest problem.
>
> [1] - Akhavan, A., Pontil, M., and Tsybakov, A. Exploiting higher order smoothness in derivative-free optimization and continuous bandits. Advances in Neural Information Processing Systems, 33:9017–9027, 2020
>
> [2] - John C. Duchi, Alekh Agarwal, Mikael Johansson, and Michael I. Jordan. Ergodic mirror descent. In 2011 49th Annual Allerton Conference on Communication, Control, and Computing (Allerton), pages 701–706, 2011.

---

> > ### Author Rebuttal · Reviewer_yMFL · 2026-04-03
> >
> > I thank the authors for their response. I will maintain my initial evaluation.

---

### Official Review · Reviewer_fkj5 · 2026-03-13

**Soundness:** 3
**Presentation:** 3
**Significance:** 2
**Originality:** 3
**Overall Recommendation:** 5
**Confidence:** 4

**Summary:**

The paper considers the problem of zeroth order stochastic optimization where the noise is markovian as opposed to i.i.d. Under this setting, the authors propose a new algorithm based on an accelerated GD method that uses a novel gradient estimator. The key contribution is the gradient estimator (two forms thereof) that is a combination of subsampling the markov chain, randomized directional gradient and MLMC. The authors provide analytical guarantees of their algorithm and show that the sample complexity of their algorithm is order optimal by establishing matching lower bounds.

**Compliance With Llm Reviewing Policy:**

Affirmed.

**Key Questions For Authors:**

Can you elaborate on the footnote 1? I had the same question but I am not sure how is it being resolved. Also how is the inherent correlation between $g_1$, $g_2$, ... being addressed in the analysis?

This might be a naive question but why does the following strategy not work: Query the same point for a while. After $O(\tau \log(1/\varepsilon))$ samples, the MC nearly converges and the samples become nearly i.i.d. Then one can use i.i.d. techniques with some error correction? Unless the error correction is quite non-trivial, this seems like an approach that would work. Given that you have established the lower bounds, it means this approach cannot be any better. I am just curious as to when this would fail and if we can further simplify the approach in this paper.

What is the role of parameter $p$? It looks like some probabilistic weighing but I am not sure why is it needed? I don't recall seeing such a parameter in regular accelerated GD with i.i.d. noise.

**Limitations:**

Yes.

**Strengths And Weaknesses:**

The paper presents an interesting approach for optimization with markovian noise. There are several key aspects in the design of the novel estimator, some of which are well known, but their combination seems interesting and useful. I think the particularly strong part is the observation of splitting into $d$ Markov chains in order to obtain optimal performance.

I don't see any glaring weakness in the paper that needs to be pointed out.

---

> ### Author Rebuttal · Authors · 2026-03-31
>
> We thank Reviewer fkj5 for their time, work and positive feedback. Next, we answer the questions raised.
>
> > Q1. (Correlations within and across batches)
>
> After applying some accelerated-SGD-esque machinery to an individual iteration of the algorithm, we arrive at the following equation:
> $\text{error}_{k+1} \leq \text{error}_k \cdot (1 - \beta) + (\text{bias of }\hat{g}) + (\text{variance of }\hat{g})$.
> We would like to bound the bias and variance of the gradient estimate, however, unlike SGD, the distribution of $g$ is not a fixed one, as it depends on the state of the chain right before the samples were collected. This is reflected in our lemma 2 - we bound these terms assuming that the state (=distribution) of the chain is arbitrary at the start of the iteration, allowing us to analyze every iteration independently.
> The within-iteration correlation of $g_1, g_2, \dots$ is addressed as described in the section 2.1 on the batching technique. Specifically, the estimates get partially decorrelated as they are applied along different directions. Then we apply the minibatch variance reduction Lemma 1 (line 233). You can find further details in Lemmas 6 and 7 (lines 980-1095) in the appendix.
>
> > Q2. (A simpler approach)
>
> If we understand correctly, the method you are describing would reduce a zero-order Markovian problem to a zero-order i.i.d. setting by replacing every “i.i.d.” oracle call with $\tilde{O}(\tau)$ underlying oracle calls. This algorithm did occur to us and its downside is the complexity penalty of times $\tau$. Whether we can do better than $\tau$ times slower is the main question behind this paper. As we established, the answer is yes, our algorithm is only $(1 + \frac{\tau}{d})$ times slower, and thus such reduction would result in a suboptimal algorithm.
>
> > Q3. (Role of parameter $p$)
>
> Momentum $p$ was first introduced in Accelerated Coordinate Descent [1, Section 5] ($1 - \beta$ there), where it was used to tackle gradient oracle with variance proportional to the norm of true gradient. Oracles with such variances occur, for example, in settings which lack access to full gradients such as coordinate descent or, as in our case, zero-order optimization - $d \| \nabla f(x) \|^2$ is introduced as part of estimator variance in Lemma 2 and is negated by a choice of $p$ on lines 1290-1295 of Theorem 1.
>
> [1] - Yu. Nesterov. Efficiency of coordinate descent methods on huge-scale optimization
> problems. SIAM Journal on Optimization, 22(2):341–362, 2012.

---

> > ### Author Rebuttal · Reviewer_fkj5 · 2026-04-03
> >
> > My questions were addressed and I would like to maintain my score as is.

---

### Official Review · Reviewer_e1d3 · 2026-03-13

**Soundness:** 2
**Presentation:** 1
**Significance:** 1
**Originality:** 2
**Overall Recommendation:** 4
**Confidence:** 4

**Summary:**

The authors consider zeroth-order optimization under markovian noise in the function evaluation. In the first-order case, markovian noise with mixing time $\tau$ yields an extra factor of $\tau$ in the convergence rate for smooth convex functions. The authors study the question of whether a similar extra multiplicative dependence on $\tau$ holds in the zeroth-order setting. They show that the answer is negative when a minibatch zeroth order (single-point) estimator is used.

The key insight is that while there is Markovian noise, in zeroth-order gradient estimation, we can consider a subchain argument where the first and $d+1$ function evaluations estimate the first coordinate gradient, the second and $d+2$ function evaluations estimate the second coordinate gradient, so on and so forth. Thus the authors show that their proposed estimator does not incur a multiplicative dependence on $\tau$ in the convergence rate, for smooth strongly convex functions.

**Compliance With Llm Reviewing Policy:**

Affirmed.

**Final Justification:**

After the rebuttal, the authors have clarified my concerns. I have thus raised my score accordingly.

**Key Questions For Authors:**

1) Why is it necessary to use minibatching in zeroth-order estimation? It seems that if no minibatching is used, the same convergence rate (in terms of number of function evaluations) can be achieved for single-point zeroth order estimation with just a single function evaluation, even with Markovian noise. In other words, we can achieve the same convergence rate (even with Markovian noise) with a much simpler algorithm, i.e. standard single-point estimator with just one function evaluation, with suitably tuned learning step size. Some discussion on this would be much appreciated.

2) It is stated in the abstract that "[the observation in the paper] provides an efficient way to interact with Markovian stochasticity: instead of invoking the expensive first-order oracle, one should use the zeroth-order oracle". However, I do not see evidence for this statement in the convergence rates of the paper. Could the authors explain more clearly what they mean by this?

3) Could the authors provide applications in optimization settings (i.e. not RL) where Markovian noise arises? This may improve the motivation of the work.

**Limitations:**

Yes.

**Strengths And Weaknesses:**

Strengths: The analysis of the minibatching argument and how estimating different directions to reduce the mixing time effect of Markovian noise is clear.

Weaknesses:

1) The organization of the paper can be improved. For instance the introduction of the central question of the paper is not very clear (right now a picture is used to introduce this question). Also, unlike most papers where there is a problem formulation section followed by main results section, here there is only a main results section. Finally, there is no conclusion section.

2) There is no discussion of why minibatching is necessary for zeroth-order estimation. Without minibatching, one can achieve the same convergence rate even with Markovian noise using just one sample in the single-point setting and two samples in the two-point setting.

---

> ### Author Rebuttal · Authors · 2026-03-30
>
> Many thanks to Reviewer e1d3 for their time, detailed review, and suggestions. We now turn to their specific concerns.
>
> > W1 (Organization of the paper)
>
> > Introduction of the central question is not very clear.
>
> The central question is formulated in the “Hypothesis” paragraph (line 155) as follows: “The hypothesis arises that the transition to zero-order Markov optimization adds two multipliers at once: $d \tau$ and $d^2 \tau/\varepsilon$ for two- and one-point.”
> While we do believe the diagram is a more intuitive way of representing this claim,
> we agree that it could be made more clear by extending it to  “<...>, resulting in $\mathcal{O}(\frac{d\tau\sigma^2}{\mu^2 \varepsilon})$ and $\mathcal{O}(\frac{d^2 \tau \sigma^2}{\mu^2 \varepsilon^2})$ complexities respectively”. We will add this clarification in the final version of the paper.
>
> > No problem formulation section.
>
> We agree, the paper presents both “Preliminaries”/”Problem formulation” and “Main Results” as a single section. We will split the combined section in the final version.
>
> > No conclusion section.
>
> Fair criticism, we will add it in the final version. For the draft, please see L1 by Reviewer hBbm.
>
> > Q1. (Same convergence rate with simpler algorithm)
>
> The core reason for the more complicated batched estimator is better convergence rate. This is because MLMC helps to control bias of stochastic "gradient" under Markov randomness (see line 280 and further + Lemma 2 on line 310 and a small discussion below). This is critically important for methods with momentum (accelerated methods). If MLMC batching is removed from the method, then convergence guarantees should worsen:
>
> - [1] considered both approaches for gradient methods (not ZO), without MLMC they achieved convergence $O(\tau^2 \cdot \frac{\tau \sigma^2}{\varepsilon^2})$ in the stochastic term, and with MLMC- batching they had $O(1 \cdot \frac{\tau \sigma^2}{\varepsilon^2})$. When preparing our paper on ZO, we also tried to analyze the method without MLMC-batching, and got the same impairments in $\tau^2$ times for the non-MLMC method.
>
> - as an alternative, we can do the following: one can wait for $\tau$ oracle calls and use only each $\tau$+1 oracle, and discard all the intermediate ones. In a sense, this brings us closer to the independent stochasticity. In this case the deterministic (first) term will degrades from $O(d \sqrt{\frac{L}{\mu}})$ to $O(\tau\cdot d\sqrt{\frac{L}{\mu}})$. From a practical point of view, it also looks strange: just skip the "gradient" calculations and wait.
>
> > Q2. (Elaborate on “efficient way to interact with Markovian stochasticity”)
>
> We believe the previous sentence of the abstract is valuable context: “Using a randomized batching scheme, we show that when mixing time $\tau$ of the underlying noise sequence is less than the dimension of the problem $d$, the convergence estimates of our method do not depend on $\tau$”. Indeed, as shown in Theorem 1, the oracle complexity scales as $(d + \tau) \sim d$ under assumption $\tau < d$, implying that Markovianity of the problem can be ignored. More broadly, assuming that evaluating a zero-order oracle is $d$ times cheaper than computing the corresponding gradient, first order methods require resources proportional to $d·\tau$ — the cost of one oracle call is $d$ and the number of calls grows linearly with $\tau$. The resource complexity of our method is instead additive: $d + \tau$.
> Both of these claims can be found in the “Computational efficiency” paragraph (line 153).
>
> > Q3. (Applications outside of RL)
>
> - We can single out statements where we deal with the black box models - do not see the weights, do not control the randomness that occur in its inference, but can evaluate the output. E.g., when we attack a proprietary/API model. One can consider the prompt tuning for API models. In more details, assume $x$ corresponds to system prompt parameters (e.g. prefix tuning), $Z$ - the stream of requests and $F(x, Z)$ is a measure of our satisfaction. $Z$ are serially correlated, as users perform repeated queries within the same context. Queries themselves may also introduce correlation, as they depend on the time-of-day \ trending events. Finally, the model may change over time as more heavily quantized versions are deployed during peak usage, contributing to short-term biases in output. If the underlying LLM is accessible as API only or $F$ is based on human feedback, the gradients become inaccessible, forcing the use of ZO.
>
> - Another direction is when we artificially introduce Markov stochasticity into the algorithm. Such approaches include, for example, the token algorithm [2], which is popular in distributed computing. In this case, any distributed ZO formulation can be solved more robustly.
>
> [1] Solodkin Methods for optimization problems with markovian stochasticity and non-euclidean geometry
>
> [2] Hendrikx A principled framework for the design and analysis of token algorithms

---

> > ### Author Rebuttal · Reviewer_e1d3 · 2026-04-04
> >
> > I thank the authors for their response to my question. My concerns have partially been resolved, and I am happy to raise my score if the authors can help to address some of my remaining questions.
> >
> > To make sure we are on the same page, I am thinking of the following "simple algorithm" for either one point or two point:
> >
> > $$x_{t+1} = x_t - \eta g_t(x_t,u_t, v_t\;Z_t)$$
> > where for the one-point estimator,
> > $$g_t(x_t,u_t,Z_t) = \frac{d (f(x_t + u_t v_t\; Z_t))}{u_t} v_t$$
> > and for the two-point estimator,
> > $$g_t(x_t,u_t,Z_t) = \frac{d \left(f(x_t + u_t v_t\; Z_t) - f(x_t - u_t v_t\; Z_t)\right)}{u_t} v_t.$$
> >
> > Above, $u_t$ denotes a smoothing radius, $v_t$ is a random vector drawn from the unit sphere, and $Z_t$ is the noise term, which may or may not be Markovian. Such one-point or two-point algorithms, that only use one or two function evaluations to make an update per iteration,  i.e. no minibatching, are quite common in the literature. See e.g. [1] for the one-point case and [2] for the two-point case. I believe these methods are applicable with or without Markovian noise.
> >
> > With that in mind, I have the following questions.
> >
> > 1) I understand that the proposed framework is beneficial primarily in the minibatch setting, to reduce the variance in the estimator at each iteration when the noise is Markovian. Having said that, why is mini-batching necessary? Is the main point of mini-batching to enable the use of accelerated scheme?
> >
> > 2) If the main purpose of mini-batching is to enable the use of accelerated scheme, what is the benefit of the accelerated algorithm with minibatching (with batch size larger than 1), and in what regime is it preferable over a standard single-point/two-point zeroth-order algorithm that uses no minibatching (i.e. batch size = 1)? It would be to good to get some clarification on this since in the stochastic setting, the noise term dominates the linear convergence term in the convergence rate. It is my understanding that the convergence rate corresponding to the noise term is identical whether minibatching is used or not.
> >
> > 3) A smaller question, but does $B$ in the paper, i.e. the batch size multiplier, mean the same thing as the batch size in the minibatch?
> >
> >
> > [1]: Flaxman et al.: "Online convex optimization in the bandit setting: gradient descent without a gradient"
> >
> > [2] Nesterov and Spokoiny: "Random Gradient-Free Minimization of Convex Functions"

---

> > > ### Author Response · Authors · 2026-04-05
> > >
> > > We thank the reviewer e1d3 for the feedback and the willingness to reconsider the score. We address the remaining questions below with additional clarifications.
> > >
> > > The "simple algorithm" you proposed is a natural approach, but it is not clear how to interpret "these methods are applicable with or without Markovian noise" in terms of the resulting convergence guarantees.
> > >
> > > For example, with 2-point feedback, we tried to prove the convergence the simple algorithm and achieve a $d \cdot \tau$ scaling in the stochastic term using analysis similar to the existing work [1-2]. Concretely, this bound can be achieved using 1-sample gradient estimators you proposed as a (black-box) first order oracle in [1] or [2] with the same mixing time $\tau$:
> > >
> > > $$\\|g_t(x_t, u_t, Z_t) - \nabla f\\|_2^2 \approx d\sigma_2^2$$
> > > and therefore the final stochastic term will be $\tau \cdot d\sigma_2^2$, i.e. not optimal $d + \tau$. And the term $d + \tau$ instead of $d \cdot \tau$ is the main contribution of the paper.
> > >
> > > It is unclear from prior work if the simple algorithm matches the optimal $\tau + d$ scaling obtained by our method. Additionally, it is not accelerated, i.e. doesn’t match our bound for small $\sigma$.
> > >
> > > [1] Even, M. Stochastic gradient descent under Markovian sampling schemes.
> > >
> > > [2] Solodkin, V., Veprikov, A., and Beznosikov, A. Methods for optimization problems with markovian stochasticity and non-euclidean geometry.
> > >
> > >
> > > > Q1. (why mini-batching)
> > >
> > > We clarify that the parameter $B$ can be chosen arbitrarily (see Theorems 1 and 1’), giving optimal $d + \tau$ scaling in the stochastic term for any $B$ and achieving optimal iteration complexity when $B = \infty$ (see remark on line 350), i.e. our framework is beneficial in all batch size regimes.
> > >
> > > However, even for $B = 1$ (i.e. $\tilde{O}(1)$ oracle calls per point on average) our estimator is not equivalent to the one-sample estimator from the "simple algorithm" you proposed, as we perform MLMC batching (see lines 280-300, second column). As you correctly mentioned, this is motivated by the standard bias reduction (see a small discussion below Lemma 2) to allow for an accelerated scheme.
> > >
> > > > Q2. (why accelerated algorithm)
> > >
> > > Pursuing the faster deterministic convergence $\frac{L}{ \mu} \to \sqrt{\frac{L}{ \mu}}$ (and, consequently, the more complex MLMC estimator) allows us to additionally obtain iteration complexity matching lower bounds even for noise-free problems ($\sigma \approx 0$). In practice the resulting algorithm would also reach the noise-dominated neighborhood ($\mathcal{O}(\sqrt{\frac{L}{ \mu}} \log \frac{1}{\varepsilon}) < \frac{(d+\tau) \sigma_2^2}{\mu^2 \varepsilon}$) of a solution faster, i.e. the linear convergence portion of the optimization would be finished in fewer iterations.
> > >
> > > You are correct that the parameter $B$ doesn’t affect the stochastic term.
> > >
> > > > Q3. (parameter $B$)
> > >
> > > Our batch size is random on each iteration with the expected batch size $\tilde{O}(B)$, so $B$ can be viewed as an average case mini-batch size parameter.

---

### Decision · Program_Chairs · 2026-04-30

**Decision:**

Accept (regular)

**Comment:**

This paper proposes a new derivative-free stochastic gradient estimator using Markov samples. Using the new MLMC sampling mechanism, the paper shows that when the Markov chain is fast mixing with ($\tau < d$), the zero-th order optimization scheme is efficient. Overall, the paper is interesting and presents a new result. After the rebuttal, the reviewers are in consensus to accept the paper. The authors are reminded to incorporate comments from the reviews in preparing the final version.